# Structure of a tetrameric photosystem I from a glaucophyte alga *Cyanophora paradoxa*

Koji Kato[1,6], Ryo Nagao [1,6✉], Yoshifumi Ueno[2], Makio Yokono [3], Takehiro Suzuki[4], Tian-Yi Jiang[1], Naoshi Dohmae [4], Fusamichi Akita [1], Seiji Akimoto [2], Naoyuki Miyazaki [5✉] & Jian-Ren Shen [1✉]

Photosystem I (PSI) is one of the two photosystems functioning in light-energy harvesting, transfer, and electron transfer in photosynthesis. However, the oligomerization state of PSI is variable among photosynthetic organisms. We present a 3.8-Å resolution cryo-electron microscopic structure of tetrameric PSI isolated from the glaucophyte alga *Cyanophora paradoxa*, which reveals differences with PSI from other organisms in subunit composition and organization. The PSI tetramer is organized in a dimer of dimers with a C2 symmetry. Unlike cyanobacterial PSI tetramers, two of the four monomers are rotated around 90°, resulting in a completely different pattern of monomer-monomer interactions. Excitation-energy transfer among chlorophylls differs significantly between *Cyanophora* and cyanobacterial PSI tetramers. These structural and spectroscopic features reveal characteristic interactions and excitation-energy transfer in the *Cyanophora* PSI tetramer, suggesting that the *Cyanophora* PSI could represent a turning point in the evolution of PSI from prokaryotes to eukaryotes.

[1] Research Institute for Interdisciplinary Science and Graduate School of Natural Science and Technology, Okayama University, Okayama 700-8530, Japan. [2] Graduate School of Science, Kobe University, Hyogo 657-8501, Japan. [3] Institute of Low Temperature Science, Hokkaido University, Hokkaido 060-0819, Japan. [4] Biomolecular Characterization Unit, RIKEN Center for Sustainable Resource Science, Saitama 351-0198, Japan. [5] Life Science Center for Survival Dynamics, Tsukuba Advanced Research Alliance (TARA), University of Tsukuba, Ibaraki 305-8577, Japan. [6] These authors contributed equally: Koji Kato, Ryo Nagao. ✉email: nagaoryo@okayama-u.ac.jp; naomiyazaki@gmail.com; shen@cc.okayama-u.ac.jp

O xygenic photosynthesis converts light energy into chemical energy and releases molecular oxygen from water, which provides the energy required for sustaining most life activities as well as oxygen needed for all aerobic life on the earth[1]. The light-driven energy conversion reactions are performed by two multi-subunit pigment-protein complexes, photosystem I and photosystem II (PSI and PSII, respectively). Among them, PSII organizes mainly into a dimer throughout photosynthetic organisms from prokaryotes to eukaryotes[2,3], whereas the structural organization of PSI is significantly diversified among the photosynthetic organisms[4–6]. While prokaryotic cyanobacteria have either trimeric[7–9] or tetrameric PSI[10–16], eukaryotic organisms possess mainly monomeric PSI[17–26]. Structures of PSI from prokaryotic to eukaryotic organisms have been solved by X-ray crystallography and cryo-electron microscopy (cryo-EM), which revealed that the main part of the PSI core is well conserved from prokaryotes to eukaryotes. However, there are differences in the subunit composition and pigment arrangement of the PSI core, reflecting the changes of the PSI core during evolution[4–26]. Importantly, most cyanobacteria do not contain trans-membrane light-harvesting complexes (LHCs), except for some specific species of cyanobacteria that contain prochlorophyte chlorophyll (Chl) *a/b*-binding proteins[27,28] and iron-stress-induced-A proteins expressed under iron-deficient conditions[29–31]. Instead, cyanobacteria mainly use water-soluble phycobilisome proteins attached to the stromal side of the membrane as their light-harvesting antennas[32]. On the other hand, the eukaryotic PSI core is surrounded by various numbers of trans-membrane LHCs[5,6], which make the PSI core a monomer and prevent it from forming trimers or tetramers. This is one of the major differences in the PSI structure between prokaryotes and eukaryotes.

*Cyanophora paradoxa* (hereafter referred to as *Cyanophora*) is a glaucophyte alga that is thought to be an ancient eukaryotic alga evolved from prokaryotes, because it has a characteristic chloroplast termed cyanelle[33]. This was corroborated by 16S and 18S rRNA-based phylogenetic analysis showing that the cyanelle is evolutionary very close to cyanobacteria[34,35]. The photosystems of cyanelles use phycobilisomes as their light-harvesting antennas and do not have trans-membrane LHCs. PSI in *Cyanophora* was first found to exist as a monomer[36], but later PSI tetramers were also found in native membranes by blue-native polyacrylamide gel electrophoresis (BN-PAGE)[10]. The structure of the tetrameric PSI core has been determined from a cyanobacterium *Anabaena* sp. PCC 7120 (hereafter referred to as *Anabaena*)[13–15], and their excitation-energy-transfer processes have also been observed[13,37]. These observations raise an interesting question as to whether the *Cyanophora* PSI tetramer has a similar structure and excitation-energy-transfer processes with those of the *Anabaena* PSI tetramer. However, the structural and excitation-energy-transfer properties of the *Cyanophora* PSI tetramer have not been reported yet.

In this study, we solved a 3.8-Å resolution structure of the PSI tetramer isolated from *Cyanophora* by single-particle cryo-EM analysis. The PSI tetramer showed unique monomer-monomer interactions entirely different from the *Anabaena* PSI tetramers[13–15]. Excitation-energy transfer of the *Cyanophora* PSI tetramer is also different from that of the *Anabaena* PSI tetramer. These results illustrate that the *Cyanophora* PSI is in the middle of a shift from oligomers to monomers in this primitive eukaryotic alga during evolution from prokaryotes to eukaryotes.

## Results

**Overall structure of the *Cyanophora* PSI tetramer.** The PSI-tetrameric cores were purified from *Cyanophora* as described in the Methods section. Biochemical and spectroscopic analyses show that this complex is functional and intact (Supplementary

Fig. 1). To determine the structure of the PSI tetramer, cryo-EM images of the wild-type PSI tetramer were obtained by a Talos Arctica electron microscope operated at 200 kV. After data processing of the resultant images by RELION (Supplementary Fig. 2, 3, and Supplementary Table 1), a final density map of the wild-type PSI tetramer was obtained with a C2 symmetry at a resolution of 4.0 Å, based on the "gold standard" Fourier shell correlation (FSC) = 0.143 criterion (Supplementary Fig. 2–6 and Supplementary Table 1). However, a dimeric form of the PSI was found in a part of the particles in the process of 3D reconstruction (Supplementary Fig. 2), which suggests that the *Cyanophora* PSI tetramer is labile during either storage of PSI or cryo-grid preparations for cryo-EM analysis.

To suppress the sample dissociation of the PSI tetramer, we employed the GraFix technique[38] to prepare the PSI tetramer, which cross-links protein subunits by glutaraldehyde before freezing the sample for cryo-EM. The cryo-EM images of the GraFix-treated, cross-linked PSI tetramer (hereafter termed GraFix PSI tetramer) were obtained by the same Talos Arctica electron microscope at 200 kV. After data processing of the resultant images by RELION (Supplementary Fig. 7, 8, and Supplementary Table 1), a final density map of the GraFix PSI tetramer was obtained with a C2 symmetry at a resolution of 3.8 Å (Fig. 1a, and Supplementary Fig. 8, 9). The cryo-EM density map of the GraFix PSI tetramer shows features of well-resolved side chains of most amino acid regions and cofactors.

The four monomers of a PSI tetramer were denoted as monomer1, monomer2, monomer1′, and monomer2′, respectively (Fig. 1b, and Supplementary Fig. 2–6). The dimeric PSI unit is organized by interactions between monomers1(1′) and 2(2′), forming a pseudo-two-fold symmetry. This reflects that the tetramer is assembled by a dimer of dimers, which are designated as monomer1/2-dimer and monomer1′/2′-dimer, respectively (Fig. 1). Examination of the subunits in the tetramer exhibits that PsaK is present in monomer1 and monomer1′ but absent in both monomer2 and monomer2′ (Fig. 1, 2). Three PSI subunits, PsaA, PsaK, and PsaL, mainly contribute to the interactions between different monomers. In particular, PsaK from one monomer is tightly associated with PsaL from the adjacent monomer at the center of the tetramer (Fig. 1b, c). These interactions have not been observed in other structures of PSI trimers and tetramers[7–9,13–16,39,40]. In particular, the *Anabaena* PSI tetramer has four PsaLs at the center and four PsaKs at the edge of the tetramer[13–15], and no interactions between PsaL and PsaK are found in the *Anabaena* PSI tetramer (Supplementary Fig. 10). In the *Cyanophora* PSI tetramer, two monomers denoted as monomer1/1′ are rotated approximately 90° relative to its counterpart in the *Anabaena* PSI tetramer, resulting in the direct interaction of PsaL of one monomer (monomer2/2′) with PsaK of the adjacent monomer (monomer1/1′) at the center of the *Cyanophora* PSI tetramer. On the other hand, at the edge of the tetramer, PsaK of one monomer (monomer2/2′) is too close to PsaL from the adjacent monomer (monomer1/1′), leading to the loss of PsaK, which occurs in monomer2 and monomer2′ (Fig. 1, and Supplementary Fig. 10). These results demonstrate that the *Cyanophora* PSI tetramer is assembled by unique interactions different from the *Anabaena* PSI tetramer.

**Structure of the PSI monomers.** For the accurate model building, we performed focused 3D classifications using masks covering each monomeric unit (monomer1 and monomer2). The final cryo-EM density maps of wild-type PSI monomer1 and monomer2 were obtained with a C1 symmetry at resolutions of 3.3 Å and 3.2 Å, respectively (Fig. 2, Supplementary Fig. 2, 3, and Supplementary Table 1). Monomer1 contains well-known eight membrane-spanning subunits (PsaA, PsaB, PsaF, PsaI, PsaJ,

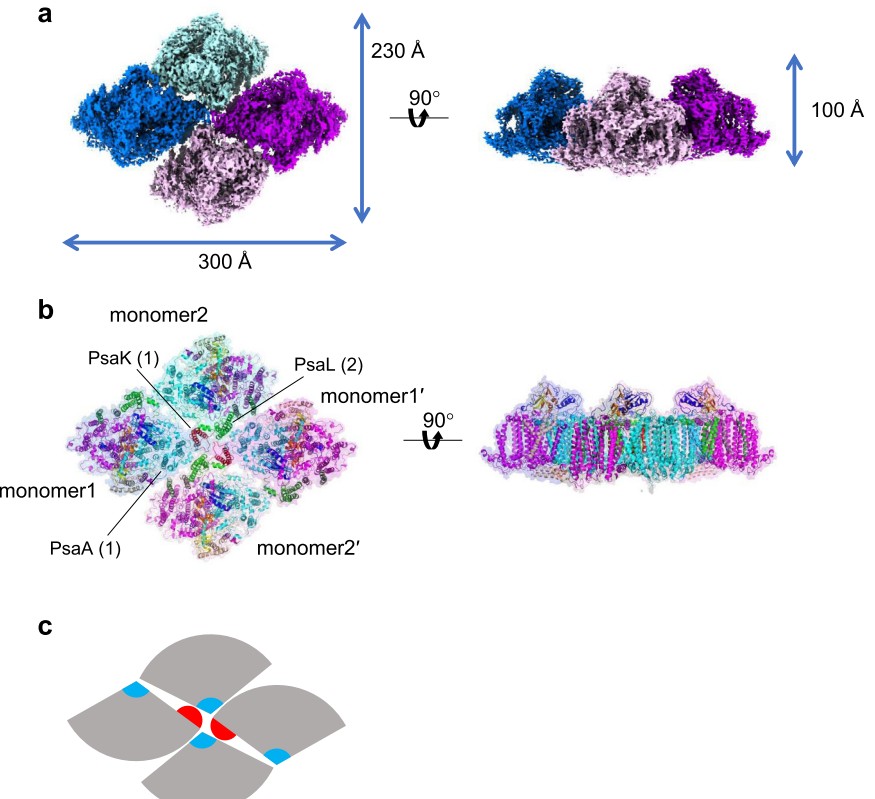

**Fig. 1 Overall structure of the GraFix PSI tetramer. a** 3D cryo-EM density map of the PSI tetramer viewed along the membrane normal from the stromal side (left) and its side view (right). **b** Structure of the PSI tetramer viewed along the membrane normal from the stromal side (left) and its side view (right). **c** Schematic diagram of the PSI tetramer. PsaK and PsaL are shown in red and cyan, respectively.

PsaK, PsaL, and PsaM) and three stromal subunits (PsaC, PsaD, and PsaE) (Fig. 2). The subunit composition of monomer2 is comparable to that of monomer1 except that PsaK is lacking. The structure of a PSI-monomer unit within the tetramer is similar to that in the cyanobacterial and plant PSI cores (Fig. 2c, d), except that the cyanobacterial PSI contains an additional subunit PsaX that is lacking in the *Cyanophora* PSI, whereas the higher plant PSI has additional PsaG and PsaH subunits but without PsaM (Fig. 2c, d). The *psaG, psaH,* and *psaX* genes are not found in the genome of *Cyanophora*[41], indicating the loss of PsaX and a PSI prior to the acquisition of PsaG and PsaH in this primitive eukaryote. There are two copies of the *psaA* and *psaB* genes in *Cyanophora,* namely, *psaA1/psaA2* and *psaB1/psaB2*; however, all of the PsaA and PsaB subunits in the tetramer structure is identified as the gene products of *psaA1* and *psaB1*. The other nine subunits have only one gene.

The cofactors identified in monomer1 and monomer2 within the tetramers are summarized in Supplementary Table 2. Monomer1 has 84 Chls *a,* 19 β-carotenes, 3 [4Fe-4S] clusters, two phylloquinones, and three lipid molecules, whereas monomer2 possesses 81 Chls *a,* 19 β-carotenes, 3 [4Fe-4S] clusters, 2 phylloquinones, and 3 lipid molecules. The locations of these molecules are similar to those in the prokaryotic and eukaryotic PSI structures[7–9,13–26], although the number of Chls differs significantly. Some of the Chls are lost from PsaB, which is likely due to the disordered structure around PsaB and/or dissociation of the cofactors during the preparation of the *Cyanophora* PSI cores.

**Interactions between monomer1 and monomer2 within a dimer.** Because of the rotation of two monomers in the tetramer (Supplementary Fig. 10), the interactions among monomers of

the *Cyanophora* PSI tetramer are clearly different from those of the *Anabaena* PSI tetramer. One interface between monomer1 and monomer2 of the *Cyanophora* PSI tetramer is formed between PsaA/K of monomer1 and PsaA/L of monomer2 at the center of the tetramer and between PsaL of monomer1 and PsaA of monomer2 at the edge of the tetramer (Fig. 3a). At the lumenal side, Asn497 of monomer1-PsaA is tightly coupled with Asn497 of monomer2-PsaA at distances of 2.5–2.6 Å (Fig. 3b). Gln88 and Ser89 of monomer1-PsaK interact with Val64, Glu65, Arg70, and Asn71 of monomer2-PsaL at distances of 2.8–3.2 Å at the center of the tetramer (Fig. 3c). Monomer1-PsaK is also associated with monomer2-PsaL through hydrophobic interactions between Ala109/Pro113 of monomer1-PsaK and Ala27/Val28 of monomer2-PsaL at the stromal side of the center (Fig. 3d). Ala27/Gly30/Leu31 of monomer1-PsaL interact with Gly317/Ile318 of monomer2-PsaA through hydrophobic interactions at the stromal side of the edge (Fig. 3e). In addition, many protein-pigment interactions are found in the monomer1-monomer2 interface.

**Interactions between monomer1 and adjacent monomer2′.** The other interface in the *Cyanophora* PSI tetramer is the interface between monomer1 and adjacent monomer2′, which is formed between PsaA of monomer1 and PsaI/L/M of monomer2′ (Fig. 4a). At the lumenal side, Chl817 of monomer1-PsaA interacts with Asn4 of monomer2′-PsaI at a distance of 2.8 Å (Fig. 4b); Chl816 of monomer1-PsaA is hydrogen-bonded to Tyr139 of monomer2′-PsaL at a distance of 3.1 Å (Fig. 4c); Tyr160/Ile164 of monomer1-PsaA interact with Phe8/BCR101 of monomer2′-PsaM through hydrophobic interactions (Fig. 4d). On the other hand, no apparent interactions are found between monomer1 and monomer2′ at the stromal side.

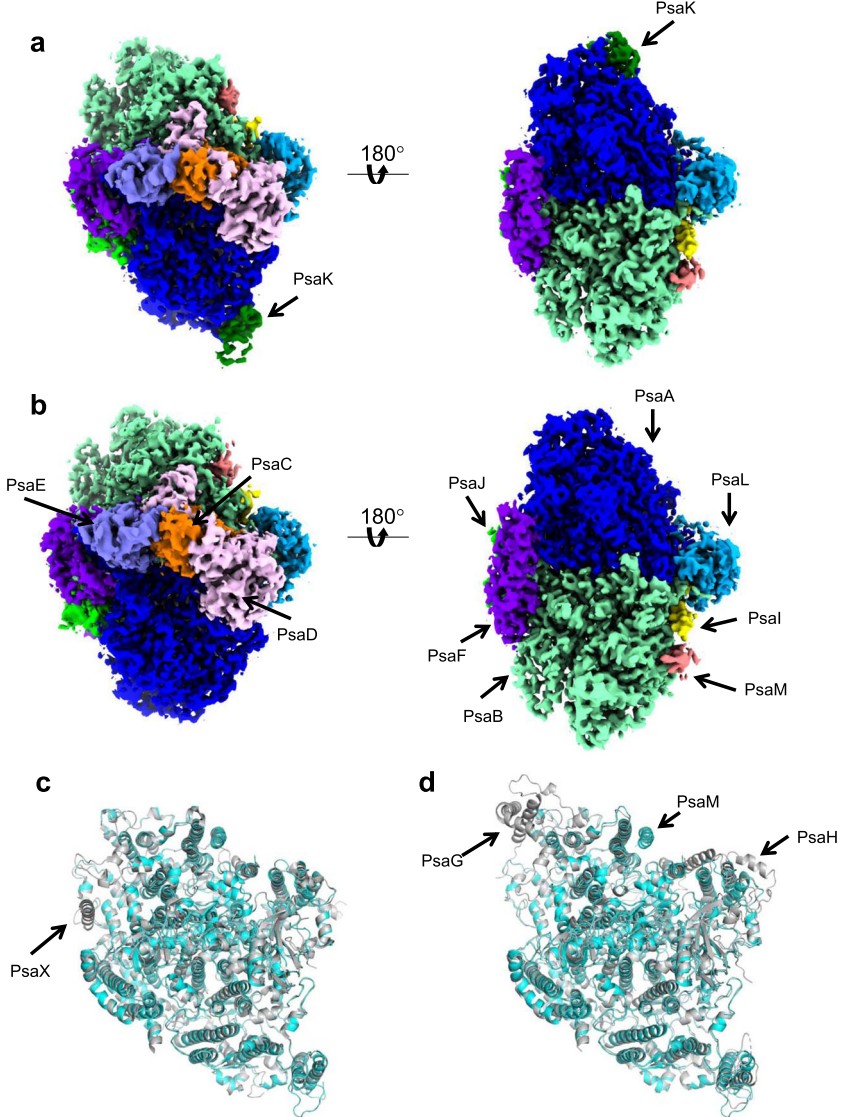

**Fig. 2 Structure of the wild-type PSI monomers. a** 3D map of monomer1 viewed along the membrane normal from the stromal (left) and lumenal (right) sides. **b** 3D map of monomer2 viewed along the membrane normal from the stromal (left) and lumenal (right) sides. **c** Superposition of the *Cyanophora* PSI monomer1 (cyan) with a PSI-monomer unit from *T. elongatus* (PDB: 1JB0) (gray), viewed along the membrane normal from the stromal side. **d** Superposition of the *Cyanophora* PSI monomer1 (cyan) with a PSI-monomer unit from *P. sativum* (PDB: 5L8R) (gray), viewed along the membrane normal from the stromal side. The subunits specific to each organism are labeled.

**Interactions at the center of the tetramer**. As mentioned above, each monomer is associated at the center of the tetramer through interactions between PsaK of one monomer and PsaL of the adjacent monomer (Fig. 5a). At the stromal side, Asn127 of monomer1(1′)-PsaK interacts with Gln129 of monomer1′(1)-PsaK at a distance of 2.8 Å (Fig. 5b). There are several complicated interactions among monomer1-PsaK, monomer2-PsaL, monomer1′-PsaK, and monomer2′-PsaL (Fig. 5c). The loop structure between Pro125 and Pro131 of monomer1-PsaK interacts with its counterpart from monomer1′-PsaK through hydrophobic interactions. Chl201 in monomer2-PsaL interacts with Pro131 of monomer1-PsaK and Phe126 of monomer1′-PsaK, and Chl201 in monomer2′-PsaL interacts with Pro131 of monomer1′-PsaK and Phe126 of monomer1-PsaK, through hydrophobic interactions.

**Pigment-pigment interactions in the tetramer**. The location of pigment molecules in monomer1 is similar to that in monomer2, except that three Chls *a* (PsaA-Chl845, PsaB-Chl835, and PsaK-

Chl201) in monomer1 are absent in monomer2 (Supplementary Fig. 11, and Supplementary Table 3). The lack of PsaB-Chl835 in monomer2 is likely due to disordered structure around PsaB or dissociation of the cofactor during the preparation of the PSI tetramer, whereas the absence of PsaA-Chl845 and PsaK-Chl201 may be due to the loss of PsaK in monomer2, because they are located at positions near PsaK in monomer1.

There are numerous pigment-pigment interactions among the monomers within the PSI tetramer (Supplementary Fig. 12a). The unique association of pigments within the tetramer are found in the interfaces at the stromal side between monomer1(1′) and monomer2(2′) but not between monomer1(1′) and monomer2′ (2). A triply stacked Chl cluster exists in the interfaces of both monomer1/2 and monomer1′/2′, which is composed of Chl823/824/846 (Supplementary Fig. 12b). The edge-to-edge distances among these Chl clusters are in the range of 3.8–4.3 Å, suggesting that these Chls may have lower energy levels. Two *β*-carotenes (BCR849 and BCR852) are close to these Chl clusters at distances of 3.6–4.6 Å (Supplementary Fig. 12b), reflecting close

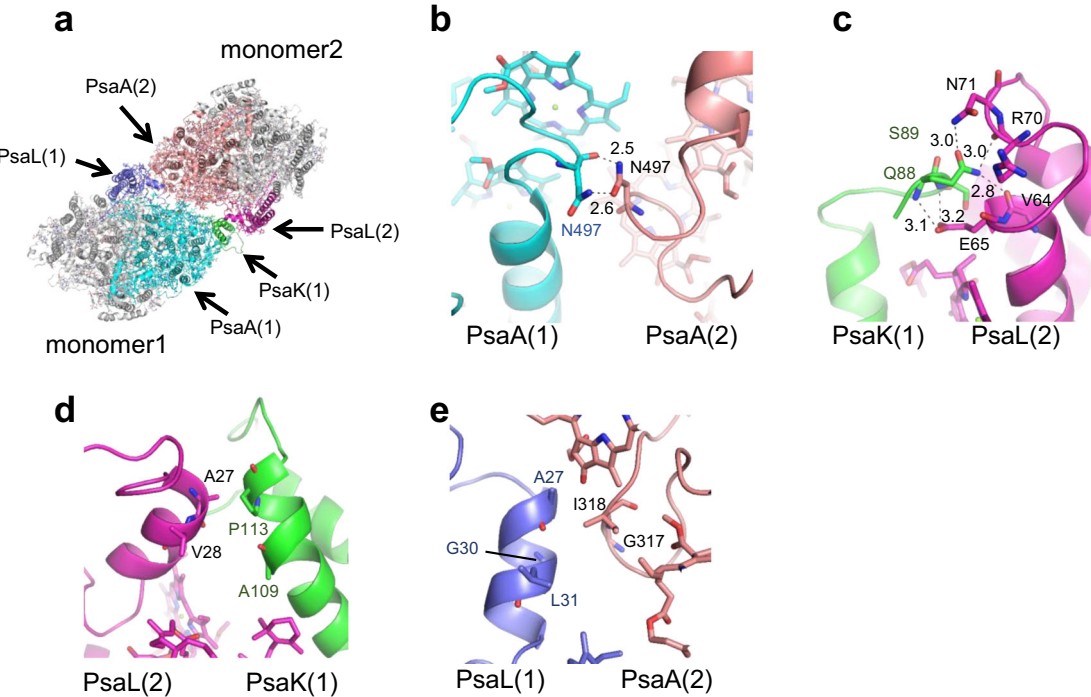

**Fig. 3 Intermonomer interactions between monomer1 and monomer2. a** Structure of a monomer1/2 dimer unit from the PSI tetramer viewed along the membrane normal from the stromal side. **b** Interactions between monomer1-PsaA and monomer2-PsaA at the lumenal side. **c, d** Interactions between monomer1-PsaK and monomer2-PsaL at the lumenal (**c**) and stromal (**d**) sides. **e** Interactions between monomer1-PsaL and monomer2-PsaA at the stromal side.

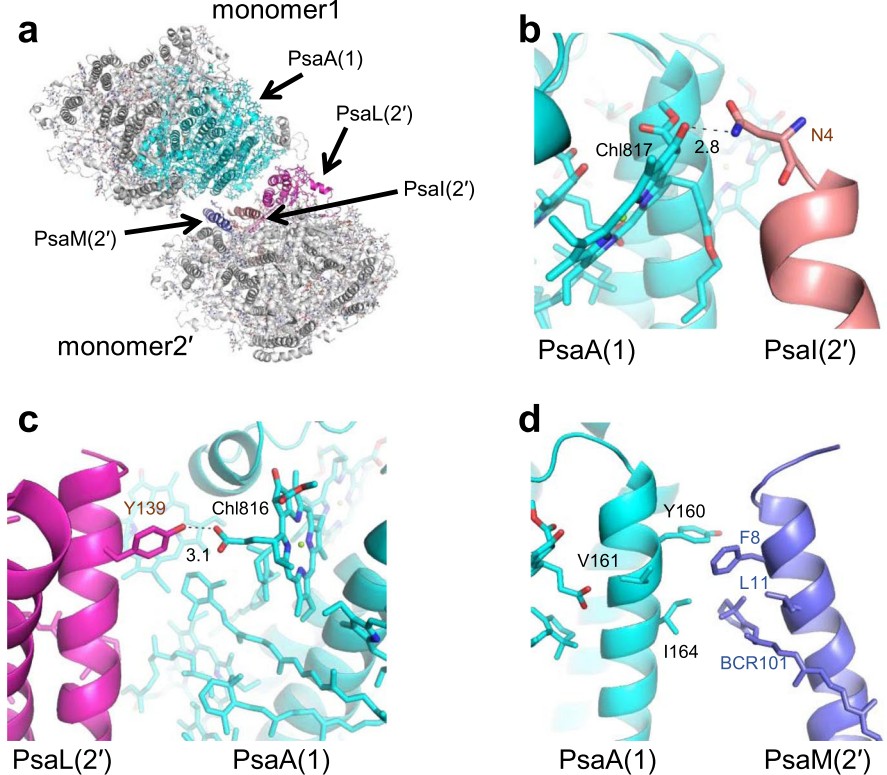

**Fig. 4 Intermonomer interactions between monomer1 and monomer2′. a** Structure of monomer1 and monomer2′ from the PSI tetramer viewed along the membrane normal from the stromal side. **b** Interactions between monomer1-PsaA and monomer2′-PsaI at the lumenal side. **c** Interactions between monomer1-PsaA and monomer2′-PsaL at the lumenal side. **d** Interactions between monomer1-PsaA and monomer2′-PsaM at the lumenal side.

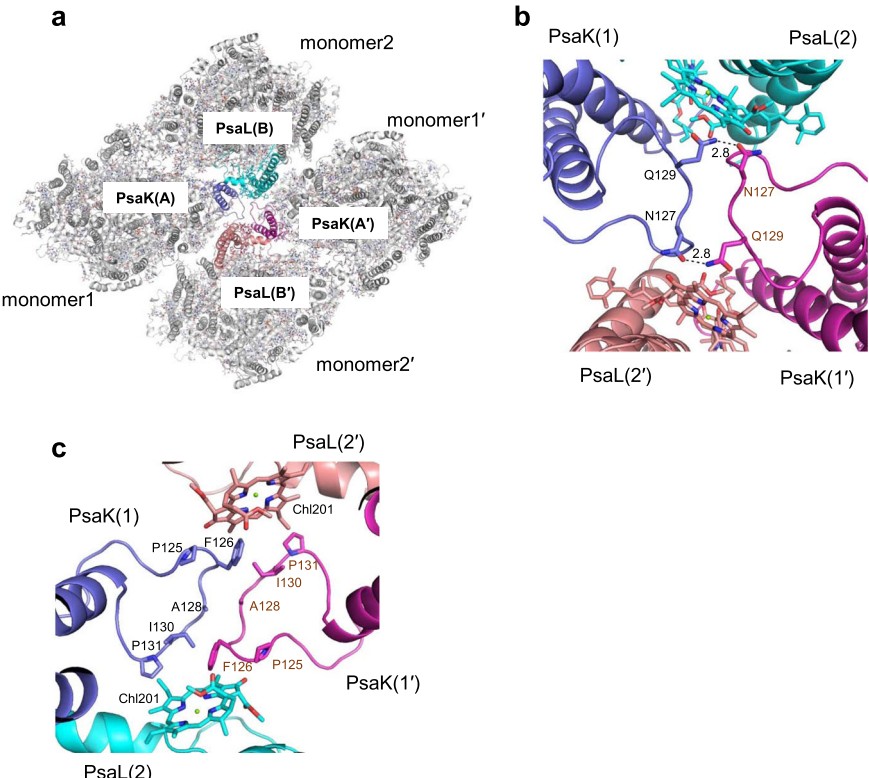

**Fig. 5 Interactions at the center of the PSI tetramer. a** Structures of the PSI tetramer viewed along the membrane normal from the stromal side. **b, c** Interactions among monomer1-PsaK, monomer2-PsaL, monomer1′-PsaK, and monomer2′-PsaL, viewed along the membrane normal from the stromal side (**b**) and viewed from the inner side of the membrane to the stromal side (**c**).

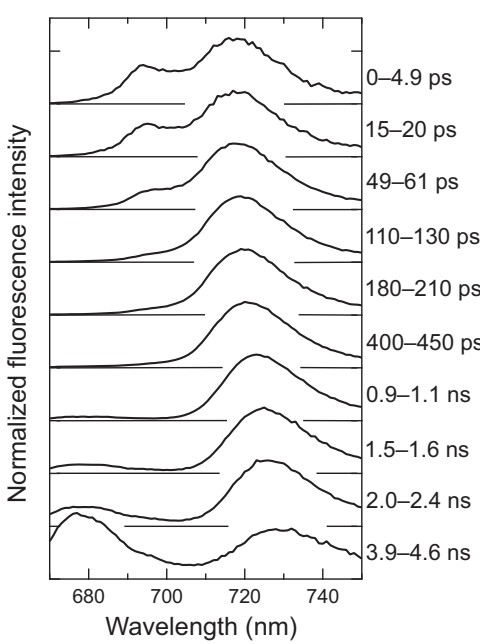

**Fig. 6 TRF spectra of the *Cyanophora* PSI tetramer.** The spectra were measured at 77 K with excitation at 445 nm, and were normalized by the maximum intensity of each spectrum.

interactions among these Chls and β-carotenes. The close Chl-β-carotene interactions may contribute to excitation-energy transfer among pigments[42–44]. In addition, PsaM-BCR101 and PsaL-BCR205 of monomer2′(2) seem to interact with PsaA-Chl808 and PsaA-Chl845 of monomer1(1′), respectively, at distances of 7.0–8.2 Å (Supplementary Fig. 12c, d). On the other hand, Chls

in different monomers are located very far among the monomers at the lumenal side, suggesting less energy transfer among the monomers at the lumenal side.

**Excitation-energy-transfer processes of the tetramer**. Time-resolved fluorescence (TRF) spectra of the PSI tetramer were measured at 77 K (Fig. 6). The TRF spectra exhibit two fluorescence bands at around 694 and 717 nm just after excitation (0–4.9 ps). Until 130 ps, the 694-nm band is lost, whereas the 717-nm band is slightly shifted to about 719 nm. This suggests excitation-energy transfer from Chls fluorescing at 694 and 717 nm to those at 719 nm, and energy trapping to the reaction-center Chls in PSI. The 719-nm band is gradually shifted to longer wavelengths to 728 nm until 4.6 ns, suggesting energy transfer to low-energy Chls fluorescing at 728 nm. In addition, fluorescence at around 677 nm is observed with a small contribution in the time range of 0.9–4.6 ns as observed for other cyanobacterial PSI preparations[13,37,45,46]. These characteristic peaks in the TRF spectra are verified in the steady-state fluorescence spectrum of the tetramer (Supplementary Fig. 1d).

## Discussion

The present study demontrates a completely different arrangement of the *Cyanophora* PSI tetramer compared with that of the *Anabaena* PSI tetramer, thereby providing a potential clue as to how the PSI-oligomerization state is determined. We compare the structures of the *Cyanophora* PSI tetramer with those of PSI trimer and tetramer in cyanobacteria (Supplementary Fig. 13a). Here we focus on the PsaL subunit because of its significant contribution to the assembly of cyanobacterial PSI[10–16,47]. Superposition of the *Cyanophora* PSI monomers with the cyanobacterial PSI trimer from *Thermosynechococcus elongatus*

(PDB: 1JB0, hereafter referred to as *T. elongatus*)[7] showed a steric hindrance by Arg45 and Ile129 of the *Cyanophora* PsaL (Supplementary Fig. 13b, 14). Ile129 is found in the *Anabaena* PsaL, whereas Val129 is found in other cyanobacteria that form PSI trimer (Supplementary Fig. 14). However, Arg45 is replaced with an uncharged residue in cyanobacteria that form either PSI trimer or tetramer (Supplementary Fig. 14). Furthermore, the C-terminus of *Cyanophora* PsaL is shorter than cyanobacterial PsaL (Supplementary Fig. 13c, 14). These factors may make the *Cyanophora* PsaL unable to form interactions required for the trimer formation. In contrast, superposition of the *Cyanophora* PSI monomers with the *Anabaena* PSI tetramer showed no steric hindrance around PsaL (Supplementary Fig. 13d–f). Interestingly, the N and C-termini of *Cyanophora* PsaL are shorter than those of *Anabaena* PsaL (Supplementary Fig. 13e, f, 14), both of which are likely important for the formation of the *Anabaena* PSI tetramer. This suggests that PsaL of *Cyanophora* cannot support the formation of a tetramer similar to the PSI structure observed in *Anabaena*.

We next examined the structure and sequence of PsaK that may differentiate *Cyanophora* from cyanobacteria and eukaryotes. An insertion between 121–131 is found in the *Cyanophora* PsaK sequence, which is absent in cyanobacteria (Supplementary Fig. 15). As described above (Fig. 5b, c), residues in this region are required for the interaction of PsaK with PsaL and PsaK from other monomers in the *Cyanophora* PSI tetramer. In addition, there are also some changes in the residues of Gln88, Ser89, Ala109, and Pro113 that interact with PsaL from the adjacent monomer (Fig. 3c, d). Among these residues, Gln88 and Ser89 are unique to the *Cyanophora* PsaK, whereas Ala109 and Pro113 are changed to different residues or are conserved in cyanobacteria (Supplementary Fig. 15). Thus, the insertion of 121–131 and changes of Gln88 and Ser89 in the sequence of *Cyanophora* PsaK may enable the formation of the atypical assembly of the PSI tetramer in *Cyanophora* and may cause rotation of two of the four monomers in the *Cyanophora* PSI tetramer relative to the *Anabaena* PSI tetramer. The insertion around residues 121–131 is also found in green algae and higher plants that form a PSI monomer. However, the sequence of this region is highly variable, and the eukaryotic PSI has the trans-membrane LHCs to surround the monomeric PSI core. Phylogenetic analyses of typical PSI subunits, PsaA, PsaB, PsaK, and PsaL, exhibit that while the *Cyanophora* PsaA, PsaB, and PsaL are either grouped within cyanobacteria or between cyanobacteria and eukaryotes, the sequences of *Cyanophora* PsaK do not group with any of the other organisms and form a unique clade consisting of *Cyanophora* only (Supplementary Fig. 16–19). This is good evidence for the uniqueness of PsaK and its role in tetramer formation in *Cyanophora*.

Despite the presence of a large hole in the center of the *Anabaena* PSI tetramer but almost no free space in the center of the *Cyanophora* PSI tetramer, the interactions of each protomer within the *Cyanophora* PSI tetramer are weaker than those within the *Anabaena* PSI tetramer. This is manifested by the fact that the GraFix method has to be used to isolate the stable *Cyanophora* PSI tetramers to prevent dissociation of the tetramer into monomers. The weak interactions among the PSI-monomer units in *Cyanophora* may be caused by significant modifications in the sequences of PsaL and PsaK, and may be helpful for dissociation of the PSI cores from oligomers to monomers. Upon complete attachment of PsaK, the tetramers will be broken and transferred to monomers. This leads us to propose a model for the oligomerization of PSI during evolution from cyanobacteria to various eukaryotes, including higher plants. Changes in the sequences of PsaL result in the cyanobacterial-type PSI tetramer, whereas changes in the PsaL and PsaK sequences result in the *Cyanophora*-type PSI tetramer seen only in the eukaryote *Cyanophora*.

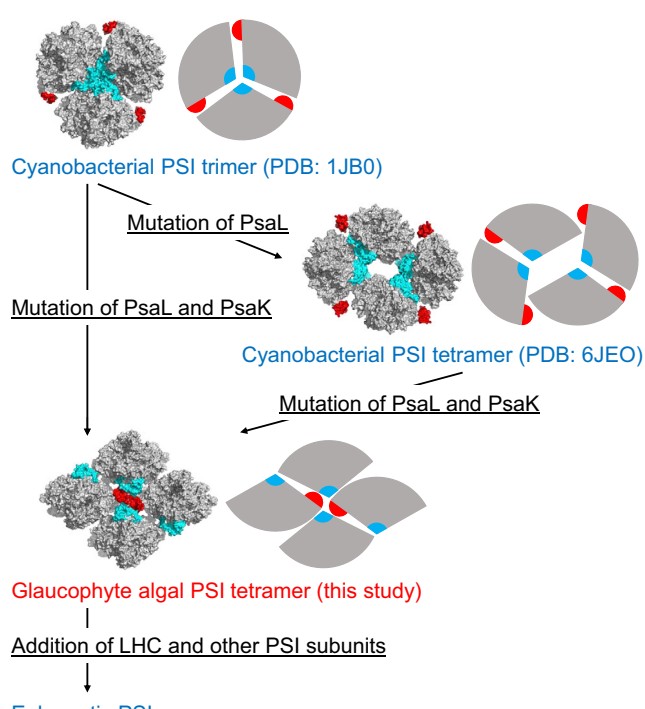

**Fig. 7 A model for the molecular evolution of PSI oligomerization state.** The structures of the PSI oligomer are shown in both surface model and schematic diagram. PsaK and PsaL are shown in red and cyan, respectively.

Furthermore, PsaH is acquired in some eukaryotes, and this subunit, together with the trans-membrane LHC subunits, may inhibit the oligomerization of PSI. Since PsaH and trans-membrane LHCs are not encoded in the *Cyanophora* genome[41], it is implied that both changes in the sequences of PsaL and PsaK and the lack of PsaH and LHCs contribute to the formation of the unusual tetrameric PSI structure in *Cyanophora*.

The above structural implications allow us to draw a model implying that *Cyanophora* is an intermediate between oxygenic photosynthetic prokaryotes and eukaryotes in the evolutionary processes of oxyphototrophs (Fig. 7). The cyanobacterial PSI cores are mainly trimers or tetramers. The changes in PsaL and PsaK induce atypical interactions responsible for the tetramerization of the *Cyanophora* PSI cores (Supplementary Fig. 13–19). For the evolution of organisms in the red and green lineages, once the *Cyanophora* PSI tetramers are dissociated into monomers, other PSI subunits and membrane-embedded LHCs may be bound to the PSI monomers. The additional bindings of PsaG, PsaH, and LHCs inhibit PSI oligomerization in the eukaryotes (Fig. 7). These observations are consistent with the evolutionary point of view that *Cyanophora* is evolved from cyanobacteria having PSI trimers and tetramers and subsequently serves as an ancestor for other eukaryotic algae where PSI becomes a monomer.

We have shown excitation-energy-transfer processes of the *Anabaena* PSI tetramer[13,37]. Different from the TRF spectra of the *Cyanophora* PSI tetramer (Fig. 6), the *Anabaena* PSI tetramer showed two fluorescence bands at around 696 and 730 nm with a shoulder at around 715 nm just after excitation (0–4.9 ps)[13]. Among the three fluorescence bands, the fluorescence at around 715 nm decreased until 130 ps in the *Anabaena* PSI tetramer but not in the PSI dimer or monomer, thereby contributing to energy transfer among PSI-monomer units by Chls fluorescing at 715 nm in the *Anabaena* PSI tetramer. In the *Cyanophora* PSI tetramer,

however, the 717-nm fluorescence band remains at 130 ps (Fig. 6). These results suggest the apparent differences in excitation-energy transfer, likely among PSI-monomer units, between *Cyanophora* and *Anabaena* PSI tetramers.

Plausible candidates of pigments for the unique energy-transfer processes in the *Cyanophora* PSI tetramer are the couplings of triply stacked Chls in the monomer1(1′)-monomer2(2′) interface and the Chl-$\beta$-carotene interactions in the interfaces between monomer1(1′) and monomer2(2′) and between monomer1(1′) and monomer2′(2) (Supplementary Fig. 12). In particular, the characteristic interactions among Chl823/824/846 (Supplementary Fig. 12) may produce low-energy Chls and characterize the time evolution of fluorescence spectra (Fig. 6). In addition, the interactions of Chls-$\beta$-carotenes, Chl823/824/846-BCR849/BCR852 (Supplementary Fig. 12), may serve as excitation-energy transfer between Chls and $\beta$-carotenes[42–44]. These unique interactions between Chls and $\beta$-carotenes would be required for the regulation of excitation energy in *Cyanophora*.

In previous studies, Koike et al. suggested that the *Cyanophora* PSI is present only in a monomeric form[36], whereas Watanabe et al. showed the presence of tetrameric, dimeric, and monomeric PSI cores in *Cyanophora*, where the amount of tetramers and dimers seemed to be decreased at lower detergent concentrations[10]. In this study, we recognized that the *Cyanophora* PSI tetramer is significantly labile. Despite the GraFix method, two additional bands of PSI complexes smaller than the PSI tetramers, presumably PSI dimers and monomers, appear in the second-round trehalose gradient centrifugation containing glutaraldehyde (Supplementary Fig. 1a). Thus, the *Cyanophora* PSI tetramers may be readily dissociated into monomers during the preparation of the PSI cores[36], consistent with the notion that the *Cyanophora* PSI is in the middle of transition from cyanobacterial trimers and tetramers to eukaryotic monomers.

The weak interactions among the PSI-monomer units in the tetramer raise the question of whether the *Cyanophora* PSI tetramer exists in vivo. We tested trehalose gradient centrifugation after solubilizing the thylakoids with a concentration of *n*-dodecyl-$\beta$-D-maltoside ($\beta$-DDM) as low as 0.1% (w/v) (Supplementary Fig. 20). The results clearly showed the existence of PSI tetramers even with the solubilization of 0.1% $\beta$-DDM. The ratio of PSI monomer to tetramer is almost unchanged between solubilization by 0.1% and 1% $\beta$-DDM. Since both PSI tetramers and monomers were observed by the BN-PAGE analysis using thylakoid membranes[10], it is suggested that the *Cyanophora* PSI exists in both tetrameric and monomeric forms in vivo. Further in situ study by cryo-electron tomography will be required for clarifying this question.

In conclusion, this study has demonstrated that the eukaryotic PSI from *Cyanophora* can form a tetrameric structure that is remarkably different from the *Anabaena* PSI tetramer. This seems to be caused by both the absence of PsaK in two of the four monomers and the unique structure of PsaL. The tetramer association of *Cyanophora* PSI is rather weak, suggesting that the tetramer can be readily dissociated into monomers. Since other photosynthetic eukaryotes have a monomeric PSI core without PSI tetramers, these structural features imply that the *Cyanophora* PSI represents an evolutionary turning-point between cyanobacteria and other photosynthetic eukaryotes.

## Methods

### Purification and characterization of the PSI tetramer from *Cyanophora*. 
The glaucophyte alga *Cyanophora paradoxa* NIES-547 was grown in 5 L of BG11 medium supplemented with 10 mM Hepes-KOH (pH 8.0) and 5 mL of KW21 (Daiichi Seimo) at a photosynthetic photon flux density of 30 μmol photons m$^{-2}$ s$^{-1}$ at 30 °C with bubbling of air containing 3% (v/v) CO$_2$. Note that KW21 is helpful for the growth of photosynthetic organisms as employed for various algae[48–50]. Thylakoid membranes were prepared after disruption of the cells with glass beads[51] and

suspended in a buffer containing 0.2 M trehalose, 20 mM Mes-NaOH (pH 6.5), 5 mM CaCl$_2$, and 10 mM MgCl$_2$. The thylakoids were solubilized with 1% (w/v) $\beta$-DDM at a Chl concentration of 0.25 mg mL$^{-1}$ for 30 min on ice in the dark with gentle stirring. After centrifugation at 20,000 × g for 10 min at 4 °C, the resultant supernatant was loaded onto a linear trehalose gradient of 10–40% (w/v) in a medium containing 20 mM Mes-NaOH (pH 6.5), 0.2 M NaCl, and 0.1% $\beta$-DDM. After centrifugation at 154,000 × g for 18 h at 4 °C (P40ST rotor; Hitachi), the PSI-tetramer fraction was obtained in a 25–30% trehalose layer and then concentrated using a 100 kDa cut-off filter (Amicon Ultra; Millipore) at 4,000 × g.

Subunit composition of the PSI tetramer was analyzed by a 16–22% SDS-polyacrylamide gel electrophoresis (PAGE) containing 7.5 M urea[52] (Supplementary Fig. 1b). The PSI tetramer corresponding to 2 μg of Chl was solubilized for 10 min at 60 °C after adding 3% lithium lauryl sulfate and 75 mM dithiothreitol. A standard molecular weight marker (SP-0110; APRO Science) was used. The subunit bands separated were identified by mass spectrometry analysis[53]. An absorption spectrum was measured at 77 K using a spectrometer equipped with an integrating sphere unit (V-650/ISVC-747; JASCO)[54] (Supplementary Fig. 1c). A steady-state fluorescence spectrum was recorded at 77 K using a spectrofluorometer (FP-8300/PMU-183; JASCO)[55] (Supplementary Fig. 1d). Pigment composition was analyzed according to the method of Nagao et al.[56,57], and the elution profile was monitored at 440 nm (Supplementary Fig. 1e).

### GraFix-treatment of the PSI tetramer. 
Initial attempts of cryo-grid preparation showed that the PSI tetramers tended to dissociate, and the GraFix method[38] was therefore used in the last centrifugation step to produce cross-linked samples for cryo-EM analysis in the presence of 0–0.05% glutaraldehyde from top to bottom in the gradient. A fraction of the tetramers was recovered, and a buffer containing 160 mM glycine, 50 mM Mes-NaOH (pH 6.5), 10 mM MgCl$_2$, 5 mM CaCl$_2$, and 0.03% $\beta$-DDM was added to stop the cross-linking reaction. The fraction was then concentrated using a 150 kDa cut-off filter (Apollo; Orbital Biosciences, USA) at 4,000 × g, with a buffer containing 50 mM Mes-NaOH (pH 6.5), 10 mM MgCl$_2$, 5 mM CaCl$_2$, and 0.03% $\beta$-DDM. The concentrated PSI tetramer was stored in liquid nitrogen until use.

### Cryo-EM data collection. 
For cryo-EM experiments, 2 μL of sample solution was applied onto a holey carbon grid (Quantifoil R2/1, Cu 300 mesh) covered with a thin amorphous carbon film. The concentrations of samples with and without cross-linking are 48 μg and 7 μg of Chl mL$^{-1}$, respectively. The grids loaded with the samples were incubated for 30 s in a chamber of an FEI Vitrobot Mark IV at 4 °C and 100% humidity. After washing with 2 μL of the solution without trehalose to increase image contrast, the grids were blotted with filter papers for 5 s and then immediately plunge-frozen into liquid ethane. The grids were examined with a 200 kV cryo-electron microscope (Talos Arctica; Thermo Fisher Scientific) incorporating a field emission gun and a direct electron detector (Falcon 3EC; Thermo Fisher Scientific). Automated data collection was performed by the EPU software (Thermo Fisher Scientific). The conditions for the cryo-EM data collection are summarized in Supplementary Table 1.

### Cryo-EM image processing. 
Cryo-EM movies were recorded at a nominal magnification of ×92,000 using the Falcon 3EC detector in a linear mode (calibrated pixel size of 1.093 Å). The movie frames were aligned and summed using the MotionCor2 software version 1.1.0[58], and the contrast transfer function (CTF) was estimated using the Gctf program version 1.18[59]. The 3D structures are reconstructed using RELION-3.0[60].

The procedure of the structural analysis is summarized in Supplementary Fig. 2. For structural analysis of the unfixed (without cross-linking) sample, in total 1,603,082 particles were automatically picked from 4,515 micrographs, and they were subjected to reference-free 2D classification. Tetramers and dimers were observed in the 2D classification. After removing bad particles in the low-resolution, good particles were further subjected to the second-round 2D classification. To determine a structure in the tetrameric form (a dimer of dimers), particles in classes viewed from the top or bottom (perpendicular to the membrane) containing only tetramers and classes viewed from the side (parallel to the membrane) probably containing tetramers and dimers were selected (1,010,216 particles) and subjected to three rounds of 3D classification with a C2 symmetry. The reference model used in the 3D classification was generated in RELION. Finally, 145,567 particles were selected and used for the 3D reconstruction. The final map in the tetrameric form was reconstructed with a C2 symmetry at 4.0 Å resolution, which was estimated by the gold-standard Fourier shell correlation at 0.143 criterion[61]. To improve the resolution in each monomeric unit, we performed focused 3D classifications using masks covering each monomeric unit. To determine higher-resolution structures in each monomeric unit (monomer1 and monomer2), we combined particles in the dimeric and tetrameric forms. Dimeric particles after the first 2D classification were selected (622,323 particles) and subjected to first-round 3D classification for the dimers. Particles in good classes were selected (416,757 particles) and combined with tetrameric particles (145,567 particles). As the tetrameric particle is a dimer of dimers and has a two-fold symmetry, the particle orientation was expanded with a C2 symmetry before joining the particles (291,134 particle orientations). The joined particle set was

subjected to the second-round 3D classification using a mask covering a dimer. Particles in good classes were selected (660,237 particle orientations) and subjected to two rounds of the focused classification using masks covering each monomeric unit. The final maps of monomer1 and monomer2 were reconstructed from 70,920 and 110,380 particles at 3.3 Å and 3.2 Å resolutions, respectively (Supplementary Fig. 3). Local resolutions were estimated using RELION (Supplementary Fig. 3).

For structural analysis of the GraFix-treated sample, in total 724,316 particles were automatically picked from 3,205 micrographs and then used for reference-free 2D classification. For the structure of the GraFix PSI tetramer, in total 426,961 particles were selected from good 2D classes and subsequently subjected to two rounds of the 3D classification with or without a C2 symmetry. The initial model used for the first 3D classification was the structure of unfixed PSI tetramer at 4.0 Å resolution. As shown in Supplementary Fig. 7c, the structure of the GraFix PSI tetramer was reconstructed from 40,679 particles at an overall resolution of 3.8 Å (Supplementary Fig. 8). Local resolution was estimated using RELION (Supplementary Fig. 8).

**Model building and refinement.** The 3.3-Å and 3.2-Å cryo-EM maps were used for the model building of the wild-type PSI monomer1 and monomer2 within the PSI tetramer, respectively. For the PSI-core model building, homology models constructed using Phyre2[62] were first manually fitted into each map with UCSF Chimera version 1.14[63], and then inspected and adjusted individually with Coot version 0.7.2[64]. The wild-type PSI monomer1 and monomer2 structures were then refined with PHENIX version 1.14 (phenix.real_space_refine)[65] with geometric restraints for protein-cofactor coordination. For structural analysis of the GraFix PSI tetramer, the wild-type PSI monomer1 and monomer2 within the tetramer were manually fitted into the 3.8-Å cryo-EM map using UCSF Chimera and were then refined with phenix.real_space_refine with geometric restraints for protein-cofactor coordinations. The final models were further validated with MolProbity version 4.4[66] and EMRinger version 1.0.0[67]. The statistics for all data collection and structure refinement are summarized in Supplementary Table 1. All structural figures are made by UCSF ChimeraX version 0.91[68] and PyMOL version 2.3.0[69].

**TRF measurement.** TRF spectra were recorded by a time-correlated single-photon counting system with a wavelength interval of 1 nm and a time interval of 2.44 ps[70]. A picosecond pulse diode laser (PiL044X; Advanced Laser Diode Systems) was used as an excitation source, and it was operated at 445 nm with a repetition rate of 3 MHz. The TRF-measurement conditions were described in detail[71].

**Phylogenetic analyses of the PsaA, PsaB, PsaK, and PsaL subunits.** Alignment and phylogenetic reconstructions were performed using the function "build" of ETE3 version 3.1.1[72] as implemented on the GenomeNet (https://www.genome.jp/tools/ete/). The tree was constructed using FastTree version 2.1.8 with the default parameters[73]. The species used for the analysis are Cyanophora paradoxa, Thermosynechococcus elongatus BP-1, Synechocystis sp. PCC 6803, Anabaena sp. PCC 7120, Chaetoceros gracilis, Chlamydomonas reinhardtii, Cyanidioschyzon merolae, Pisum sativum, and Zea mays.

**Statistics and Reproducibility.** Numerous PSI particles were picked up from the cryo-EM images and used for structural analysis with standard protocols. The data statistics and evaluation of the resolution were documented in Supplementary Fig. 2, 3, 7, 8, and Supplementary Table 1.

**Reporting Summary.** Further information on research design is available in the Nature Research Reporting Summary linked to this article.

## Data availability
The data that support this study are available from the corresponding authors upon reasonable request. Atomic coordinates and cryo-EM maps for the reported structures of monomer1, monomer2, and the GraFix-treated PSI tetramer have been deposited in the Protein Data Bank under the accession codes 7DR0, 7DR1, and 7DR2, and in the Electron Microscopy Data Bank under the accession codes EMD-30820, EMD-30821, and EMD-30823, respectively. The cryo-EM map of the wild-type PSI tetramer has also been deposited in the Electron Microscopy Data Bank under the accession code EMD-30822. Source data are provided with this paper.

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

## Acknowledgements

We thank Drs. Naruhiko Adachi and Masato Kawasaki for helpful assistance during the cryo-EM study. This work was supported by the Platform Project for Supporting Drug Discovery and Life Science Research (Basis for Supporting Innovative Drug Discovery and Life Science Research (BINDS)) from AMED, JSPS KAKENHI grant Nos. JP20H02914 (K.K.), JP20K06528, JP21K19085 (R.N.), JP18J10095 (Y.U.), JP19K22396, JP20H03194 (F.A.), JP16H06553 (S.A.), JP20H05087 (N.M.), and JP17H06433 (J.-R.S.), Takeda Science Foundation (K.K.), and TIA-Kakehashi grant No. TK19-048 (N.M.).

## Author contributions

R.N. and J.-R.S. conceived the project; K.K., R.N., and T.-Y.J purified the PSI cores and performed biochemical characterizations; Y.U., M.Y., and S.A. performed spectroscopic measurements and analyzed the data; T.S. and N.D. identified PSI subunits by mass spectrometry analysis; K.K. performed phylogenetic analyses; F.A. and N.M. collected cryo-EM images; K.K. and N.M. processed the EM data and reconstructed the final EM maps; K.K. built the structure model and refined the final models; K.K., R.N., S.A., N.M., and J.-R.S. wrote a draft manuscript; and R.N. and J.-R.S. wrote the final manuscript, and all of the authors joined the discussion of the results.

## Competing interests

The authors declare no competing interests.
