## [Peer Review File · Nature Communications]

Structure of a tetrameric photosystem I from a glaucophyte alga *Cyanophora paradoxa*Reviewers' Comments:

Reviewer #1:

Remarks to the Author:

Kato et al. describe the structure of a tetrameric photosystem I (PSI) complex purified from the alga *Cyanophora paradoxa*. Structures of *Cyanophora*'s photosynthetic complexes are desirable and interesting because it is thought that an endosymbiotic event led to the unique chloroplasts found in this organism. Unlike other algae whose PSI binds peripheral antenna proteins similar to those found in plants, *Cyanophora*'s PSI binds phycobilisomes for antenna like that observed in cyanobacteria. Thus, the cyanobacteria-like PSI found in *Cyanophora*, an alga, has relevance to understanding the evolution of photosynthesis.

In cyanobacteria, PSI is found in oligomeric states of trimers, tetramers, and monomers. In plants, PSI exists only as monomers and are surrounded by LHC peripheral antenna proteins. The first report of *Cyanophora* PSI's oligomeric state found only monomeric PSI. The second showed a population of tetrameric PSI at very high detergent concentration. Because some other cyanobacteria are known to form tetrameric PSI, the finding that *Cyanophora* exhibit tetrameric PSI was of interest.

Here, Kato et al. isolated PSI from *Cyanophora* and present (a) a 4-angstrom resolution structure of the PSI in a tetrameric organization, (b) a 3.8-angstrom resolution structure of cross-linked tetrameric PSI, and (c, d) structures of the PSI monomers that comprise the dimer-of-dimers in the tetramer where focused refinement allowed for higher resolution, around 3.2 angstroms.

The tetrameric arrangement observed in the structure is surprisingly dissimilar from that observed in other tetrameric PSI structures from cyanobacteria. In two of the monomers, the PsaK subunit is absent, which allows for the tetrameric arrangement. The authors suggest various other unique interactions, including some that implicate the PsaL subunit which is known to impact the oligomeric state of PSI.

The authors use the structure to suggest an evolutionary scheme for PSI oligomerization suggesting that *Cyanophora* PSI tetramerization is representative of an ancestor of plant and algal PSI that appeared shortly after the split of PSI that could produce trimers and tetramers.

While I think the structure itself is interesting, I disagree with the assumptions made regarding the physiological relevance of the *Cyanophora* PSI tetramer, and I find the evolutionary implications to be highly speculative. Thus, I cannot recommend this manuscript for publication in *Nature Communications* in its current state. I describe my major and minor concerns below. In brief, for the manuscript to be relevant, the authors would need to show that tetrameric PSI exists *in vivo*, and perform a more thorough phylogenetic analysis that supports their evolutionary scheme.

Major: There is little evidence that tetrameric PSI in *Cyanophora* is biologically relevant.

The tetrameric oligomeric state of *Cyanophora* PSI is established by only a single paper, that from Watanabe et al. which the authors cite. The only other work analyzing the oligomeric state of *Cyanophora* PSI is from Koike et al. (2000) where only monomeric PSI was detected from size exclusion chromatography. This suggests a discrepancy in the literature as to whether tetrameric PSI is relevant.

Watanabe et al. screened the oligomeric state of *Cyanophora* PSI from 0.8-5% DDM, which is about 20-200 times higher than that commonly used. In that work, it was clear that the tetrameric state of detergent-solubilized *Cyanophora* PSI increases with DDM concentration, as the authors here also mention. This is in contrast to the better-characterized PSI tetramer from *Anabaena* which disassembles

in high detergent as is common for protein complexes. In Cyanophora, there is also a small population of PSI dimers appearing at the lowest concentration screened by Watanabe et al., 0.8% DDM. There is little discussion about this trend in Watanabe et al., but to me, it very strongly suggests that the tetrameric PSI formation is an artifact of detergent solubilization. In other words, the population of PSI tetramers may not be present in the absence of detergent (i.e. in vivo).

The authors here use 0.1% DDM for solubilization which is considerably less than the concentrations screened in Watanabe et al. (0.8 – 5.0%). They see a large population of dimeric PSI in their particle data set, suggesting that the tetrameric form is quite unstable which the authors also mention. This is consistent with the idea that the tetrameric state is an artifact of detergent solubilization. Since neither Koike et al. or Watanabe et al. saw PSI dimers at 0.8% detergent, and there are many dimers in the sample here at 0.1% detergent, it suggests that as detergent increases, monomers form dimers and eventually tetramers, probably as PsaK is lost, and thus may not be physiologically relevant.

At line 132, the authors state that “PsaK of one monomer is too close to PsaL from the adjacent monomer, leading to its loss”, however since there are more tetramers formed as detergent is increased, it can be inferred that the detergent is causing the removal of PsaK which is allowing for the formation of the tetrameric complex, making the PSI tetramer physiologically irrelevant.

To prove relevance, the authors must show that the tetrameric state of PSI is present in Cyanophora in vivo.

Major: The GraFix procedure appears to increase particle heterogeneity.

Another concern is the justification for the GraFix PSI tetramer structure. The authors state that the GraFix procedure was used to “suppress sample dissociation of the PSI tetramer”, but only about 6% of the picked particles ended up in the final GraFix tetramer data set whereas about 9% of the picked particles ended up in the native tetramer data set. Furthermore, the authors use about 150,000 particles from 4,515 micrographs for the native tetramer, therefore about 33 particles per micrograph were used in the final 3D reconstruction, but only about 40,000 particles from 3,200 micrographs for the GraFix tetramer, therefore about 12 particles per micrograph were used in the final 3D reconstruction. Both of these observations suggest that the number of useful tetramer particles collected from the GraFix data is less than that for the native data, implying greater instability in the GraFix PSI. These numbers are even more surprising because, according to the methods, the authors applied about seven times more GraFix PSI to the grids compared to native PSI.

The GraFix reconstruction did lead to a slightly higher resolution 3D reconstruction using less particles, but it is important to recognize that while this does imply an increase in stability, such is only present in a very small number of particles. Overall, it appears that the GraFix PSI was actually more heterogeneous than the native PSI. To me, the authors could show that GraFix suppresses sample dissociation by simply reporting the number of particles selected in obviously tetrameric 2D classes, and report that as a percentage of total particles picked. In any case, it sounds like there were major problems stabilizing the PSI tetramer, which, again, suggests that tetramers may be an artifact of detergent solubilization.

Major: The evolutionary points are unconvincing and cursory.

It is important to note that evolutionary analyses usually start with sequence comparisons that provide a basis for structural observations. Here, all the evolutionary arguments are essentially made based on minor structural features, usually steric hindrance. What sequence-based observations are made do not significantly support the unique tetrameric formation. While the evolution of photosynthesis in Cyanophora is interesting because it likely arose from an endosymbiotic event, I could find very little literature on the evolution of their photosystems. The authors make suggestions about evolution based on oligomeric state and the presence or absence of subunits, but without phylogenetic data, the structural observations are arbitrary and speculative.

The authors could easily enhance this aspect of the manuscript by, for example, by creating a few phylogenetic trees based on the sequences of the subunits. For example, I made a small tree from the core polypeptide PsaA and PsaB sequences (attached). This type of analysis can be used in the following way: The authors state that PsaX was lost. Because Cyanophora core polypeptides and *T. elongatus* core polypeptides cluster closely, and because the *psaX* gene is found in other cyanobacteria, it indeed suggests that PsaX was lost in Cyanophora.

On the other hand, the authors also claim that PsaG and PsaH are “pre-gained” (though I would suggest rewording that term), which I think means that Cyanophora represents a state prior to the appearance of PsaG and PsaH. If this is true, one would expect Cyanophora PSI core polypeptides to share some sequence identity unique to higher plant and algal PSI in the regions that interact with PsaG and PsaH. The authors do not address this, and based on the phylogenetic tree I provided I would expect that no sequence identity exists in Cyanophora PSI that is unique to core residues near PsaG and PsaH in plant/algal PSI, which speaks against the author’s statement regarding these subunits.

If the authors decide to enhance their evolutionary arguments in the way I describe, I would imagine they would include various other core polypeptide sequences and of course the other PsaA and PsaB isoforms. They could also make the same effort with PsaL and PsaK since these, too, are implicated in the formation of the PSI tetramer reported.

The PsaL sequence alignment (Figure S11) shows that the sequence of PsaL from Cyanophora compared to the two cyanobacteria are extremely similar except that the two terminal regions are truncated. However, more sequences should be included to support (or refute) the evolutionary conclusions. I have

included more sequences in the alignment below. While the Arg residue noted by the authors does not appear to be conserved in the alignment, in *Anabaena*, an Arg is found in the next amino acid position. Furthermore, the sequence I found for *T. elongatus* PSI also has this Arg (though it seems to be a different sequence than that used by the authors). For the Ile noted by the authors, this residue is conserved in *Anabaena* and is Val in *Chlamydomonas* and Ile or Leu in *T. elongatus*, all of which are similarly-sized hydrophobic residue. In *P. sativum*, this residue is the larger hydrophobic Phe. Regardless, these sequence alignment-based observations provided by the authors do not substantially provide evidence that PsaL in *Cyanophora* contributes to tetramerization.

While it is somewhat interesting that the termini are truncated, this is reminiscent of plant and algal PSI, which exhibit monomers only. This also supports the idea that the tetrameric state reported here is an artifact of detergent solubilization.

P. sativum _PsaL	-----KPTYQVIQPIINGDPFFIGSLETPTVT	24
C. reinhardtii _PsaL	MAVAMRSSTGLRATAARRQMPLGLGRVSTVVRVCAADTKKAQVISPVNGDPFVGMLETPTVT	60
Cyanophora _PsaL	-----MAKDAVKPFYDDAFIGHLSTPIS	23
T. elongatus _PsaL	-----MAQDVIANGGTPEIGNLATPIN	22
Anabaena _PsaL	-----MAQAVDASKNLPSPDRNREVVFAGRDPQWGNLETPTVN	38
	: : * * * * .	
P. sativum _PsaL	SSPLIAWYLSNLPAYRTAVSPLLRLGIEVGLAHGYLLVGPFFVKAGPLRNTE-IAGQAGSLA	83
C. reinhardtii _PsaL	SAPIVATYLSNLPAYRTGVAPVLRGVEIGLAHGFLLAGPFIKLGPLRNVPETAETAGSLS	120
Cyanophora _PsaL	NSSAVNGLLANLPAYRKGLTPRLRGLIEGMAHGYPFLTGPFFVELGPLRNTE-GGILYGSLS	82
T. elongatus _PsaL	SSPFRTRTFINALPIYRRGLSSNRRGLEIGMAHGFLLYGPPFSILGPLRNTE-TAGSAGLLA	81
Anabaena _PsaL	ASPLVKWFINNLPAYRPGLTPFRRGLEVGMAHGYPFLGPFPAKLGPLRDAA-NANLAGLLG	97
	: : * * * * . : : * * * * * * * * . . * *	
P. sativum _PsaL	AGGLVVILSLCLTIYGISSFNEGAPSTAPSLTLTGRKKEPDQLQTADGWAKFTGGFFFGG	143
C. reinhardtii _PsaL	AAGLVLILALCLSIYGSAQFQSTPSI--GVKTLSGRSVARDPLFSADGWSEFAAGFLVGG	178
Cyanophora _PsaL	AVGLVVILTAACLALYGFANFSGSS-----KSKDATLWESGEGWSDFVSGWLLIGG	131
T. elongatus _PsaL	TVGLVVILTVCLSLYGNAGSGPSAAE----STVTTPNPPQELFTKEGWSEFTSGFILGG	136
Anabaena _PsaL	AIGLVVLFTLALSLYANS--NPPTAL----ASVTVPNPPDAFQSKEGWNNFASAFLLIGG	150
	: * * * * : : . * * * . : : * * * * * * * * . * *	
P. sativum _PsaL	ISGVIWAYFLLYVLDLPY-----	161
C. reinhardtii _PsaL	EAGVAWAYVCTQILPYYS-----	196
Cyanophora _PsaL	AGSVGFAYLLLQYIL-----	146
T. elongatus _PsaL	LGGAFFAFYLASTPY-VQPLVKIAAGVWSVH	166
Anabaena _PsaL	IGGAVVAYFLTSNLALIQLVGV-----	172
	. . . * :	

Observations are also made regarding PsaK. I think the authors should include more sequences in the alignment as with PsaL, however PsaK is among the least conserved subunits in PSI and organisms often contain multiple isoforms of this subunit. Additionally, in the *T. elongatus* PSI structure, sidechains of PsaK were not resolved and it is therefore difficult to be confident about PsaK structural comparisons. I assume this is why there is little focus on PsaK when discussing the structural basis for tetramerization in *Cyanophora* around line 251. Since tetramer formation here implicates PsaK, there should be a stronger focus on PsaK. As with the PsaL analysis, I am not convinced that the residues the authors point out in Figure S12 really confer tetramerization. This is especially true because the loss of PsaK seems to result in tetramerization.

Minor

There are some problems with grammar and sentence structure throughout the manuscript, especially in the Introduction and Discussion.

The authors should mention how many isoforms of PsaK are present in Cyanophora.

Supplementary Figure 1: There are no arrows on panel A.

Reviewer #2:

Remarks to the Author:

In this manuscript the authors describe and characterize a tetrameric PSI complex purified from a single celled eukaryotic alga, *Cyanophora paradoxa*, that belongs to an ancient branch in the evolution of photosynthetic eukaryotes. Typically, eukaryotic PSI complexes are not seen forming higher oligomers, this manuscript has the potential to provide important insights regarding the transition from the prokaryotic trimeric and tetrameric complexes to the monomeric complexes usually observed in algae and higher plants. In general, this manuscript provides new data and merits publication in nature communications provided that the authors can put some concerns to rest. The first and most important point is the biological relevance of the identified tetrameric arrangement. This is a critical issue and the authors do very little to support the notion that this form of PSI exists in cells. In supplementary figure 1 the authors present a density gradient which contains a significant amount of tetramer, they do not specify if this is the GraFix gradient or not and they should. If this gradient contains a cross linker then a gradient without one should be shown, ideally the results from a few independent isolation procedures should be presented. A critical factor seems to be the presence of the PsaK subunit which has to be lost from one monomer, while this can clearly be an artifact of the isolation procedure, it is also common for large complexes to exist in several states, the authors should address these issues in their discussion. What type of evidence supports the differential association of PsaK in the two PSI forms? Can they show or cite data on the regulation of PsaK levels in *Cyanophora paradoxa*? In addition, the local resolution maps in supplementary figure 3d suggest that certain parts of PsaB may be disordered. The authors also state that some chlorophylls were lost from PsaB when they compare the pigment composition between the cyanobacterial PSI and the current structure. The authors should clearly name the missing pigments and examine the sequence of PsaB around these positions, if the coordinating side chains are present then I would think this would be a strong indication that further optimization is needed for the isolation procedure. From supplementary figure 8, some chlorophylls are also lost around PsaF, the authors should again correlate this with sequence changes as a way to support the native state of the complex. The authors present the 77 K emission obtained from the tetramer. Given the concerns that I raised above, they should compare purified monomers to tetramers and examine any changes in absorbance and emission. In particular they should show the room temp unnormalized emissions from PSI monomers and tetramers (in similar ODs) to see if there are any large changes in the quantum efficiency of the complex, which again can support the notion that the tetramer occurs in cells or at the very least is functional with regards to energy transfer.

With regards to the data analysis the main issue is the poor distribution of views in the final datasets. The authors should quantitate this using cryoEF or a similar software and see if this is a limiting factor in their data. In their FSC graphs the authors do not present their phase randomized correlations and these should be supplied together with the masks used in the focused refinement and classification which should be shown as part of the data analysis flow. The map examples provided by the authors are not sufficient, they should readily resolve side chains at 3.2 Å resolution and none of their map samples show this. upon resubmission they should provide the full experimental map and model together with appropriate map examples in their supplement file.

In summary, the results presented in this manuscript are potentially very interesting but require careful verification, and as much support as can be gained for the existence of this new tetrameric arrangement of PSI in cells. The manuscript should also be edited for language, I've listed a few examples below under minor points.

Minor points:

Abstract line 2: needs to be rephrased.

Line 59: the sentence should be rephrased to indicate whether the authors mean any trans membrane antennae in cyanobacteria, which ignores the presence of IsiA. Or whether they refer to LHC's only.

Line 150 – "indicating the loss of PsaX and pre-gaining of PsaG" – "and" should be removed.

Lines 269-271: "Upon further transition *Cyanophora* to other eukaryotes, PsaH, and LHCs appeared, which make PSI as a monomer (Figs. 2d and 6)." This sentence needs to be rephrased.

Modifications and our responses to the comments by Reviewer #1

Comment 1:

Kato et al. describe the structure of a tetrameric photosystem I (PSI) complex purified from the alga *Cyanophora paradoxa*. Structures of *Cyanophora*'s photosynthetic complexes are desirable and interesting because it is thought that an endosymbiotic event led to the unique chloroplasts found in this organism. Unlike other algae whose PSI binds peripheral antenna proteins similar to those found in plants, *Cyanophora*'s PSI binds phycobilisomes for antenna like that observed in cyanobacteria. Thus, the cyanobacteria-like PSI found in *Cyanophora*, an alga, has relevance to understanding the evolution of photosynthesis.

In cyanobacteria, PSI is found in oligomeric states of trimers, tetramers, and monomers. In plants, PSI exists only as monomers and are surrounded by LHC peripheral antenna proteins. The first report of *Cyanophora* PSI's oligomeric state found only monomeric PSI. The second showed a population of tetrameric PSI at very high detergent concentration. Because some other cyanobacteria are known to form tetrameric PSI, the finding that *Cyanophora* exhibit tetrameric PSI was of interest.

Here, Kato et al. isolated PSI from *Cyanophora* and present (a) a 4-angstrom resolution structure of the PSI in a tetrameric organization, (b) a 3.8-angstrom resolution structure of cross-linked tetrameric PSI, and (c, d) structures of the PSI monomers that comprise the dimer-of-dimers in the tetramer where focused refinement allowed for higher resolution, around 3.2 angstroms.

The tetrameric arrangement observed in the structure is surprisingly dissimilar from that observed in other tetrameric PSI structures from cyanobacteria. In two of the monomers, the PsaK subunit is absent, which allows for the tetrameric arrangement. The authors suggest various other unique interactions, including some that implicate the PsaL subunit which is known to impact the oligomeric state of PSI.

The authors use the structure to suggest an evolutionary scheme for PSI oligomerization suggesting that *Cyanophora* PSI tetramerization is representative of an ancestor of plant and algal PSI that appeared shortly after the split of PSI that could produce trimers and tetramers.

While I think the structure itself is interesting, I disagree with the assumptions made regarding the physiological relevance of the *Cyanophora* PSI tetramer, and I find the evolutionary implications to be highly speculative. Thus, I cannot recommend this manuscript for publication

in Nature Communications in its current state. I describe my major and minor concerns below. In brief, for the manuscript to be relevant, the authors would need to show that tetrameric PSI exists in vivo, and perform a more thorough phylogenetic analysis that supports their evolutionary scheme.

Author reply 1:

First of all, we thank the reviewer for his/her careful reading and valuable comments and suggestions, which were found important in improving our manuscript. According to the reviewer's comments and suggestions, we modified our manuscript which are listed in the following. In brief, we performed a more thorough phylogenetic analysis of various subunits of PSI, and found that PsaK of Cyanophora forms a unique clade different from other organisms, among the photosynthetic organisms examined. This makes PsaK bind to PSI loosely only in Cyanophora, resulting in the easy loss of some of the PsaK subunits. This easy loss of PsaK is never found in cyanobacteria and other eukaryotes, and thus is a unique feature of the Cyanophora PSI. As our structure shows, the loss of two PsaK subunits among the four PSI monomers is necessary for the formation of the PSI tetramer. The easy loss of PsaK in the Cyanophora PSI is consistent with the formation of a tetramer in this organism, and may suggest that PSI is in a transition state from trimer/tetramer to monomer. In the native state, PsaK is bound to PSI, which will then form a monomer in this cyanobacterium. Upon loss of the two PsaK subunits, it forms a tetramer whose structure is different from the typical tetramer found in other cyanobacteria. This is why PSI is found mainly as a monomer, but some tetramer is also found upon increasing the detergent concentration to solubilize it. We hypothesize that these tetramers are in a transition state to the monomers, and are present in vivo in a small amount. They are unstable in nature, as has indeed been observed in the present study, where we found that the tetramer has to be cross-linked in order to obtain a stable, higher-resolution structure. These results suggest that the tetramer may be an assembly intermediate or pre-mature form of the monomer in this particular organism. It has to be pointed out that the β -DDM concentration we used here to solubilize the thylakoid membrane is 0.1%, which is not high as compared with those used in other organisms, yet it gives rise to a fraction of the tetramers. All these results suggest a transit state of PSI from trimer/tetramer in cyanobacteria to monomer in eukaryotes. We have modified the manuscript by incorporating these points.

Comment 2:

Major: There is little evidence that tetrameric PSI in Cyanophora is biologically relevant. The tetrameric oligomeric state of Cyanophora PSI is established by only a single paper, that from Watanabe et al. which the authors cite. The only other work analyzing the oligomeric state

of Cyanophora PSI is from Koike et al. (2000) where only monomeric PSI was detected from size exclusion chromatography. This suggests a discrepancy in the literature as to whether tetrameric PSI is relevant.

Watanabe et al. screened the oligomeric state of Cyanophora PSI from 0.8-5% DDM, which is about 20-200 times higher than that commonly used. In that work, it was clear that the tetrameric state of detergent-solubilized Cyanophora PSI increases with DDM concentration, as the authors here also mention. This is in contrast to the better-characterized PSI tetramer from *Anabaena* which disassembles in high detergent as is common for protein complexes. In Cyanophora, there is also a small population of PSI dimers appearing at the lowest concentration screened by Watanabe et al., 0.8% DDM. There is little discussion about this trend in Watanabe et al., but to me, it very strongly suggests that the tetrameric PSI formation is an artifact of detergent solubilization. In other words, the population of PSI tetramers may not be present in the absence of detergent (i.e. in vivo).

The authors here use 0.1% DDM for solubilization which is considerably less than the concentrations screened in Watanabe et al. (0.8 – 5.0%). They see a large population of dimeric PSI in their particle data set, suggesting that the tetrameric form is quite unstable which the authors also mention. This is consistent with the idea that the tetrameric state is an artifact of detergent solubilization. Since neither Koike et al. or Watanabe et al. saw PSI dimers at 0.8% detergent, and there are many dimers in the sample here at 0.1% detergent, it suggests that as detergent increases, monomers form dimers and eventually tetramers, probably as PsaK is lost, and thus may not be physiologically relevant.

At line 132, the authors state that “PsaK of one monomer is too close to PsaL from the adjacent monomer, leading to its loss”, however since there are more tetramers formed as detergent is increased, it can be inferred that the detergent is causing the removal of PsaK which is allowing for the formation of the tetrameric complex, making the PSI tetramer physiologically irrelevant.

To prove relevance, the authors must show that the tetrameric state of PSI is present in Cyanophora in vivo.

Author reply 2:

Thanks for this important comment. Koike et al. (2000) indeed reported PSI monomer from Cyanophora by size exclusion chromatography, and we cited this paper in our modified manuscript. However, in that study, the size exclusion chromatography was performed for a

fraction of PSI isolated from sucrose density gradient centrifugation, and there was no data showing the pattern of the sucrose density gradient centrifugation in that paper. It is possible that the PSI tetramer is present in a small amount in the sucrose density gradient, and was not recovered by that study. The latter study by Watanabe et al. (2011) clearly showed the presence of tetramers in *Cyanophora*, which is confirmed in the present study.

The DDM concentration that Watanabe et al. used is indeed high compared to those used in other studies. However, the present study used 0.1% DDM to solubilize the membrane which is very low compared with those used in other studies, and we still obtained a fraction of PSI tetramers. At this concentration, PsaK is never dissociated from PSI in cyanobacteria and other eukaryotes; thus, we consider that the easy loss of PsaK and formation of the PSI tetramer are inherent features of PSI in this organism. The PSI tetramer may be an assembly intermediate or pre-mature form of PSI monomer; upon full attachment of PsaK, it will transform to monomers. This characterizes PSI in this organism as a transit state from trimer/tetramer in cyanobacteria to monomer in eukaryotes. These have been added and discussed in the modified manuscript.

As the reviewer pointed out, a certain amount of dimer was observed on the cryo-EM grid. However, as shown in Fig. S1a (unfixed), no dimer was observed during the sample purification, and a dimer-like band was confirmed only in the GraFixed sample (Fig. S1a, fixed). We should point out that the samples loaded onto the grid originally are from the tetramer band. Thus, the dimer observed on the cryo-EM grid is considered to be ones dissociated from the tetramer during the grid preparation. Therefore, there is no reason to suggest that the PSI tetramer is an artifact of detergent solubilization.

As stated above, since we used a very low concentration of detergent to obtain the tetramer in this study, we did not consider that the tetramer is an artifact induced by the detergent treatment. PsaK is not found to be lost easily in cyanobacteria and other eukaryotes, but some of the PSI do not contain PsaK in this organism only. Thus, we consider rather that the PSI tetramer is an assembly intermediate or pre-mature form in which, PsaK is not associated. The *Cyanophora* PSI transforms from cyanobacterial-type trimer/tetramer to eukaryotic-type monomer by making dissociation of PsaK easily from PSI and therefore form a unique tetramer structure different from the cyanobacterial-type tetramer. This in turn points to an important issue of the assembly process of PSI, and these points are added into the revised manuscript.

To conclude, the PSI tetramer was obtained at a very low concentration of detergent in the present study, and has a unique structure different from the conventional cyanobacterial-type

tetramer structure. The tetramer is formed by dissociation of a part of PsaK, which was never observed in cyanobacteria and other eukaryotic organisms. Combining with the unique evolutionary status of Cyanophora, these results suggest that the tetramer exists in vivo in a small amount and PSI in this organism uses this strategy to transform from cyanobacterial-type tetramer to eukaryotic-type monomer.

Comment 3:

Major: The GraFix procedure appears to increase particle heterogeneity.

Another concern is the justification for the GraFix PSI tetramer structure. The authors state that the GraFix procedure was used to “suppress sample dissociation of the PSI tetramer”, but only about 6% of the picked particles ended up in the final GraFix tetramer data set whereas about 9% of the picked particles ended up in the native tetramer data set. Furthermore, the authors use about 150,000 particles from 4,515 micrographs for the native tetramer, therefore about 33 particles per micrograph were used in the final 3D reconstruction, but only about 40,000 particles from 3,200 micrographs for the GraFix tetramer, therefore about 12 particles per micrograph were used in the final 3D reconstruction. Both of these observations suggest that the number of useful tetramer particles collected from the GraFix data is less than that for the native data, implying greater instability in the GraFix PSI. These numbers are even more surprising because, according to the methods, the authors applied about seven times more GraFix PSI to the grids compared to native PSI.

The GraFix reconstruction did lead to a slightly higher resolution 3D reconstruction using less particles, but it is important to recognize that while this does imply an increase in stability, such is only present in a very small number of particles. Overall, it appears that the GraFix PSI was actually more heterogeneous than the native PSI. To me, the authors could show that GraFix suppresses sample dissociation by simply reporting the number of particles selected in obviously tetrameric 2D classes, and report that as a percentage of total particles picked. In any case, it sounds like there were major problems stabilizing the PSI tetramer, which, again, suggests that tetramers may be an artifact of detergent solubilization.

Author reply 3:

We agree with the reviewer that GraFix increased the heterogeneity of the sample, and there were much less particles picked up in the GraFix-fixed samples. However, as shown in supplementary Fig. 6, the structures of native and GraFix PSI tetramers are very similar. With

GraFix PSI, the resolution has improved a bit and the map quality has improved significantly. Using this GraFix PSI electron microscope map, it became possible to determine the interaction between PSI monomers (supplementary Fig. 6 and 9). Supramolecular complexes *in vivo* are often unstable, which makes structural analysis difficult. This is the reason we used the GraFix PSI samples. The instability of the PSI tetramer pointed out by the reviewer cannot be the reason for the artifact of the PSI tetramer. GraFix is randomly immobilized by chemical cross-linking of the primary amine of the protein with glutaraldehyde, which is thought to have increased the heterogeneous complexes. As shown in the added supplementary Fig. 1a, a band of PSI dimer not seen in the unfixed sample was observed in the GraFix-fixed sample, which also indicates the heterogeneity of the sample.

Comment 4:

Major: The evolutionary points are unconvincing and cursory.

It is important to note that evolutionary analyses usually start with sequence comparisons that provide a basis for structural observations. Here, all the evolutionary arguments are essentially made based on minor structural features, usually steric hindrance. What sequence-based observations are made do not significantly support the unique tetrameric formation. While the evolution of photosynthesis in Cyanophora is interesting because it likely arose from an endosymbiotic event, I could find very little literature on the evolution of their photosystems. The authors make suggestions about evolution based on oligomeric state and the presence or absence of subunits, but without phylogenetic data, the structural observations are arbitrary and speculative.

The authors could easily enhance this aspect of the manuscript by, for example, by creating a few phylogenetic trees based on the sequences of the subunits. For example, I made a small tree from the core polypeptide PsaA and PsaB sequences (attached). This type of analysis can be used in the following way: The authors state that PsaX was lost. Because Cyanophora core polypeptides and *T. elongatus* core polypeptides cluster closely, and because the *psaX* gene is found in other cyanobacteria, it indeed suggests that PsaX was lost in Cyanophora.

On the other hand, the authors also claim that PsaG and PsaH are “pre-gained” (though I would suggest rewording that term), which I think means that Cyanophora represents a state prior to the appearance of PsaG and PsaH. If this is true, one would expect Cyanophora PSI core polypeptides to share some sequence identity unique to higher plant and algal PSI in the regions

that interact with PsaG and PsaH. The authors must address this to support their evolutionary conclusions.

If the authors decide to enhance their evolutionary arguments in the way I describe, I would imagine they would include various other core polypeptide sequences and of course the other PsaA and PsaB isoforms. They could also make the same effort with PsaL and PsaK since these, too, are implicated in the formation of the PSI tetramer reported.

The PsaL sequence alignment (Figure S11) shows that the sequence of PsaL from Cyanophora compared to the two cyanobacteria are extremely similar except that the two terminal regions are truncated. However, more sequences should be included to support (or refute) the evolutionary conclusions. I have included a few more sequences in the alignment below. While the Arg residue noted by the authors does not appear to be conserved in the alignment, in *Anabaena*, an Arg is found in the next amino acid position. Furthermore, the sequence I found for *T. elongatus* PSI also has this Arg (though it seems to be a different sequence than that used by the authors). For the Ile noted by the authors, this residue is conserved in *Anabaena* and is Val in *Chlamydomonas* and Ile or Leu in *T. elongatus*, all of which are similarly-sized hydrophobic residue. In *P. sativum*, this residue is the larger hydrophobic Phe. Regardless, these sequence alignment-based observations provided by the authors do not substantially provide evidence that PsaL in Cyanophora has evolved to result in tetramerization.

While it is somewhat interesting that the termini are truncated, this is reminiscent of plant and

algal PSI, which exhibit monomers only. This also supports the idea that the tetrameric state reported here is an artifact of detergent solubilization.

```

P.sativum_PsaL      -----KPTYQVIQPINGDPFIGSLETPVT 24
C.reinhardtii_PsaL  MAVAMRSSTGLRATAARRQMPGLGLGRVSTVRVCAADTKKAQVISPVNGDPFVGMLETPVT 60
Cyanophora_PsaL     -----MAKDAVKPFYDDAFIGHLSTPIS 23
T.elongatus_PsaL    -----MAQDVIANGGTPEIGNLATPIN 22
Anabaena_PsaL       -----MAQAVDASKNLPSPDRNREVVFAPGRDPQWGNLETPVN 38
                    : : * * * :.

P.sativum_PsaL      SSPLIAWYLSNLPAYRTAVSPLLRGIEVGLAHGYLLVGPVFKAGPLRNTE-IAGQAGSLA 83
C.reinhardtii_PsaL  SAPIVATYLSNLPAYRTGVAPVLRGVEIGLAHGFLLAGPFIKLGPLRNPETABIAGSLS 120
Cyanophora_PsaL     NSSAVNGLLANLPAYRKGLTPRLRGLIEIGMAHGYFLTGPVFLGPLRNTD-GGILYGSLS 82
T.elongatus_PsaL    SSPFTRTFINALPIYRRGLSSNRGLEIGMAHGFLLYGPFISILGPLRNTE-TAGSAGLLA 81
Anabaena_PsaL       ASPLVKWFINNLPAYRPGTTPFRRGLEVGMAGHYFLFGPFAKLGPLRDAANANLAGLLG 97
                    : : * * * .: : * *: *: *: *: * * * * * :. . * *.

P.sativum_PsaL      AGGLVVILSLCLTIYGISSFNEGAPSTAPSLTLTGRKKEPDQLQTADGWAKFTGGFFFGG 143
C.reinhardtii_PsaL  AAGLVLILALCLSIYGSAQFQSTPSI--GVKTLSGRSVARDPLFSADGWSEFAAGFLVGG 178
Cyanophora_PsaL     AVGLVVILTACLALYGKANFSGSS-----KSKDATLWESGEGWSDVFSVGLWIG 131
T.elongatus_PsaL    TVGLVVILTCLSLYGNAGSGPSAAE----STVTPNPPQELFTKEGWSEFTSGFILGG 136
Anabaena_PsaL       AIGLVVLFLLSLYANS--NPPTAL----ASVTVPNPPDAFQSKEGWNNFASAFILGG 150
                    : * * * : : . * : * . : : * * . * . . . . * *

P.sativum_PsaL      ISGVIWAYFLLYVLDLPY----- 161
C.reinhardtii_PsaL  EAGVAWAYVCTQILPYYS----- 196
Cyanophora_PsaL     AGSVGFAYLLQYIL----- 146
T.elongatus_PsaL    LGGAFFAFYLASTPY-VQPLVKIAAGVWSVH 166
Anabaena_PsaL       IGGAVVAYFLTSNLALIQLVGV----- 172
                    ... *:

```

Observations are also made regarding PsaK. I think the authors should include more sequences in the alignment as with PsaL, however PsaK is among the least conserved subunits in PSI and organisms often contain multiple isoforms of this subunit. Additionally, in the *T. elongatus* PSI structure, sidechains of PsaK were not resolved and it is therefore difficult to be confident about PsaK structural comparisons. I assume this is why there is little focus on PsaK when discussing the structural basis for tetramerization in *Cyanophora* around line 251. Since tetramer formation here implicates PsaK, there should be a stronger focus on PsaK. As with the PsaL analysis, I am not convinced that the residues the authors point out in Figure S12 really confer tetramerization. This is especially true because the loss of PsaK seems to result in tetramerization.

Author reply 4:

Thanks for this important comment. 18S and 16S rRNA-based phylogenetic and morphological phylogenetic analyzes have shown that glaucophyte diverged earliest among eukaryotic photosynthetic organisms. We added the sentence “This was corroborated by 16S and 18S rRNA-based phylogenetic analysis showing that the cyanelle is evolutionary very close to cyanobacteria^{28, 29}” on lines 68-70 of page4. We considered the process of molecular evolution of PSI from these reports, analysis of phylogenetic data of more subunits and organisms, as well as the structure of the PSI tetramer determined in the present study as follows.

According to the reviewer's comments, we created some phylogenetic trees based on the sequences of some PSI subunits (following figures) by incorporating sequences from more organisms. Analysis of the phylogenetic tree by amino acid sequences showed that the *Cyanophora paradoxa* in the core subunits of PsaA and PsaB were also the earliest among eukaryotic photosynthetic organisms. On the other hand, phylogenetic analysis of PsaK shows that only *Cyanophora* is divided into a completely different, unique branch, while other organisms follow the order from cyanobacteria to eukaryotes. These results suggest that special changes occurred in the sequences of PsaK in the cyanophora, which allowed it to detach easily from PSI and made PSI to form tetramers. Interestingly, diatoms do not have PsaK, and it is re-appeared in the green algae and higher plants. On the other hand, the sequences of PsaL are poorly conserved, and therefore phylogenetic analysis of PsaL does not seem to make sense. PsaG contributes to the interaction with the LHC and does not appear to be involved in the oligomerization of PSI. The interaction partner of PsaH is mainly PsaL. As mentioned above, PsaL has a wide variety and is not very well conserved. In other words, PsaL would need to be changed at the same time as acquiring PsaH. According to the comment of the reviewer, we changed the word "pre-gained" to "before the acquisition of PsaG, PsaH" in the modified text (line 150, page 7).

According to the comments of the reviewer, we included various organisms in analyzing the phylogenetic trees of PsaA, PsaB, and also added the phylogenetic trees of PsaL and PsaK, into the revised manuscript. These figures are attached below.

Regarding the involvement of PsaL in the tetramer formation, the reviewer is right that Ile129 of PsaL is changed to the same Ile or similar Val residue in other cyanobacteria, and thus is not responsible for the tetramer formation. Arg45 is unique in the *Cyanophora* PsaL, and in other cyanobacteria it is an uncharged, hydrophobic residue. More important, the C-terminus of *Cyanophora* PsaL is much shorter than PsaL of other cyanobacteria. These may contribute to the failure of cyanobacterial-type tetramer formation in *Cyanophora*. The truncation of the C-termini of the PsaL in higher plants also suggest that it cannot form the cyanobacterial-type tetramer. The formation of monomer in higher plants and green algae is due to the surrounding of the PSI core by trans-membrane LHC subunits, which are added into the revised manuscript. We are not sure why the sequences of PsaL of *T. elongatus* the reviewer used are somewhat different from the one we used here, but we confirmed that our sequences are correct for *T. elongatus*.

We increased the sequences of PsaK (and PsaL) in the phylogenetic analysis according to the reviewer's comments, which are listed below, and put more focus on PsaK, as the release of PsaK in part of monomers is the main force for PSI to form tetramers in Cyanophora. As stated above, PsaK form a unique clade in Cyanophora different from the other cyanobacteria and eukaryotes. This reinforces that PsaK is easily detached only in Cyanophora, which makes PSI a tetramer in this organism only. We added these into the revised manuscript.

Minor

Comment 5:

There are some problems with grammar and sentence structure throughout the manuscript, especially in the Introduction and Discussion.

The authors should mention how many isoforms of PsaK are present in Cyanophora.

Author reply 5:

Thank you for the comments and suggestions. We checked the grammar throughout the text, and correct ones where necessary. We added the sentence “The other nine subunits have only one gene.” on lines 153-154 of page7.

Comment 6:

Supplementary Figure 1: There are no arrows on panel A.

Author reply 6:

We added arrows on panel A; thank you.

Modifications and our responses to the comments by Reviewer #2

Comment 1:

In this manuscript the authors describe and characterize a tetrameric PSI complex purified from a single celled eukaryotic alga, *Cyanophora paradoxa*, that belongs to an ancient branch in the evolution of photosynthetic eukaryotes. Typically, eukaryotic PSI complexes are not seen forming higher oligomers, this manuscript has the potential to provide important insights regarding the transition from the prokaryotic trimeric and tetrameric complexes to the monomeric complexes usually observed in algae and higher plants. In general, this manuscript provides new data and merits publication in nature communications provided that the authors can put some concerns to rest.

Author reply 1:

First of all, we thank the reviewer for his/her highly positive and encouraging comments. According to the reviewer's comments, we modified the manuscript as follows.

Comment 2:

The first and most important point is the biological relevance of the identified tetrameric arrangement. This is a critical issue and the authors do very little to support the notion that this form of PSI exists in cells. In supplementary figure 1 the authors present a density gradient which contains a significant amount of tetramer, they do not specify if this is the GraFix gradient or not and they should. If this gradient contains a cross linker then a gradient without one should be shown, ideally the results from a few independent isolation procedures should be presented.

A critical factor seems to be the presence of the PsaK subunit which has to be lost from one monomer, while this can clearly be an artifact of the isolation procedure, it is also common for large complexes to exist in several states, the authors should address these issues in their discussion. What type of evidence supports the differential association of PsaK in the two PSI forms? Can they show or cite data on the regulation of PsaK levels in *Cyanophora paradoxa*?

Author reply 2:

Thanks for this important comment. According to the reviewer's comment, we replaced Fig. S1a by the following figure. In this new panel, left side shows the density gradient of unfixed (uncross-linked, original one) sample and right side one shows the GraFixed (cross-linked) sample, respectively. Red arrows are PSI tetramers.

As the reviewer pointed out, PsaK subunit has to be lost from two monomers in order to form the PSI tetramer. While this could be an artifact upon detergent solubilization, we used 0.1% DDM which is very low in comparison with those used in solubilizing the membranes of other cyanobacteria and eukaryotes. In these cyanobacteria and eukaryotes, PsaK was not found to be detached specifically even under much higher concentrations of detergent-treatment. Furthermore, phylogenetic analysis showed that PsaK of Cyanophora groups into a unique clade different from those of cyanobacteria and other eukaryotes, while other subunits (PsaA, PsaB, PsaL) do not (Supplementary Figs. S16-S19, revised manuscript). All these results suggest that PsaK is unique in Cyanophora, and the tetrameric PSI may be an assembly intermediate or pre-mature form before transition to the PSI monomer, but not an artifact. We added these points to the revised manuscript.

We were unable to provide data on the regulation of PsaK levels in the cyanophora paradoxa, and there were no such reports so far. We may do so in the future to confirm the level of PsaK. However, giving the low amount of PSI tetramer, the level of PsaK may not be much different from those of other subunits, and it might be difficult to gain insight from the level of PsaK. On the other hand, we will analyze the membrane with cryo-electron tomography to check whether the Cyanophora PSI complexes exist in tetramers and monomers in vivo.

Comment 3:

In addition, the local resolution maps in supplementary figure 3d suggest that certain parts of PsaB may be disordered. The authors also state that some chlorophylls were lost from PsaB when they compare the pigment composition between the cyanobacterial PSI and the current structure. The authors should clearly name the missing pigments and examine the sequence of PsaB around these positions, if the coordinating side chains are present then I would think this would be a strong indication that further optimization is needed for the isolation procedure. From supplementary figure 8, some chlorophylls are also lost around PsaF, the authors should again correlate this with sequence changes as a way to support the native state of the complex.

Author reply 3:

We apologize for making a mistake in the handedness shown in the Figs S3d and e. The disordered part of Fig. S3d is PsaA. This part has a flexible structure without interaction with surrounding molecules (Fig. 1a). Also, in the PSI tetramer, PsaB is always located outside the complex. According to the reviewer's comments, we added a label to Fig. S8 and summarized the ligands of Chl a and compared them with *T. elongatus* in Supplemental Table S3. As can be seen in the Table, most of the lost Chl is bound to the molecular surface of PsaB, and the ligands for these lost Chls are all water molecules in *T. elongatus* structure which are not visible in the Cyanophora structure. PsaB has many disordered parts and its structure itself is flexible, and it is difficult to solubilize and purify it to determine its structure. In the future, we will analyze the native structure of the Cyanophora PSI complex in vivo by the cryo-electron tomography method.

Comment 4:

The authors present the 77 K emission obtained from the tetramer. Given the concerns that I raised above, they should compare purified monomers to tetramers and examine any changes in absorbance and emission. In particular they should show the room temp unnormalized emissions from PSI monomers and tetramers (in similar ODs) to see if there are any large changes in the quantum efficiency of the complex, which again can support the notion that the tetramer occurs in cells or at the very least is functional with regards to energy transfer.

Author reply 4:

We agree with the reviewer's comment. However, we are unable to purify PSI monomer because of contamination of phycobiliproteins in the PSI-monomer-enriched fraction of density gradient centrifugation. So, we cannot measure the absorbance and fluorescence spectra of PSI

monomer at present. Nevertheless, the small differences in the pigment composition between monomer and tetramer, even if present, would make little differences in the absorption and fluorescence emission spectra. We will measure the absorption and fluorescence spectra of monomer and compare them with the tetramer in the future when we can obtain pure monomers from *Cyanophora*.

Comment 5:

With regards to the data analysis the main issue is the poor distribution of views in the final datasets. The authors should quantitate this using cryoEF or a similar software and see if this is a limiting factor in their data. In their FSC graphs the authors do not present their phase randomized correlations and these should be supplied together with the masks used in the focused refinement and classification which should be shown as part of the data analysis flow. The map examples provided by the authors are not sufficient, they should readily resolve side chains at 3.2 Å resolution and none of their map samples show this. upon resubmission they should provide the full experimental map and model together with appropriate map examples in their supplement file.

Author reply 5:

Thanks for this important comment. The Efficiency (EOD) calculated by CryoET was 0.72 for the GraFixed tetramer. This is not a bad value. Following the reviewer's comments, we added an example of phase randomized FSC and map to the supplementary figures (Supplementary Figs 3-5, 8 and 9).

Comment 6:

In summary, the results presented in this manuscript are potentially very interesting but require careful verification, and as much support as can be gained for the existence of this new tetrameric arrangement of PSI in cells. The manuscript should also be edited for language, I've listed a few examples below under minor points.

Author reply 6:

We thank the reviewer for his/her highly positive and encouraging comments again. We believe that the unique PSI tetramer revealed in the present study represents a transition form from cyanobacterial-type trimer/tetramer to eukaryotic-type monomer, and is present in vivo in *Cyanophora*. According to the reviewer's comments, we have carefully checked the language throughout the text and correct ones where necessary. The points raised by the reviewer is corrected as follows.

Minor points:

Comment 7:

Abstract line 2: needs to be rephrased.

Author reply 7:

We replaced to the sentence “The oligomerization state of PSI is variable depending on the species of organisms.” as suggested (p2, lines 27-28, modified manuscript).

Comment 8:

Line 59: the sentence should be rephrased to indicate whether the authors mean any trans membrane antennae in cyanobacteria, which ignores the presence of IsiA. Or whether they refer to LHC’s only.

Author reply 8:

We added the sentence “except for the antenna protein (*isiA*), which is expressed during iron deficiency²²⁻²⁴,” in page3, Line 58.

Comment 9:

Line 150 – “indicating the loss of PsaX and pre-gaining of PsaG” – “and” should be removed.

Author reply 9:

We modified the original sentence to “indicating the loss of PsaX and before the acquisition of PsaG” (line 150, page 7, revised manuscript).

Comment 10:

Lines 269-271: “Upon further transition Cyanophora to other eukaryotes, PsaH, and LHCs appeared, which make PSI as a monomer (Figs. 2d and 6).” This sentence needs to be rephrased.

Author reply 10:

We replaced to the sentence with: “Eukaryotes other than *Cyanophora* have acquired PsaH, and this subunit, together with the trans-membrane LHC subunits, inhibit the oligomerization of PSI and therefore make them a monomer (Figs. 2d and 6).”, as suggested by the reviewer (p12, lines 296-298).

Reviewers' Comments:

Reviewer #1:

Remarks to the Author:

Although I think the manuscript is much improved, the authors have not addressed my main concern. As I stated previously, "To prove relevance, the authors must show that the tetrameric state of PSI is present in *Cyanophora* in vivo."

My main criticism is that there is no data suggesting that tetramers are present in vivo, and there is data that shows correlation between DDM and this unique tetramer state. The most straightforward interpretation is that the tetrameric state the authors have isolated is not physiologically relevant, and is instead induced by DDM.

The authors state that 0.1% DDM is very low for solubilization, which I agree with, but the standard procedure is to solubilize at high DDM concentration, separate the protein target, and then bring the DDM concentration to ~0.02%, which is around 2 times the CMC of DDM. Here, the authors solubilize the sample at 0.1% and then maintain the 0.1% which is around 20 times the CMC of DDM. Why this deviation from standard practice? I assume this is because 0.02% DDM results in too few tetramers for cryo-EM which would be consistent with the results from Watanabe that show a decrease in tetramers with lower DDM concentration.

Reviewer #2:

Remarks to the Author:

My concerns regarding the functionality and relevance of the PSI tetramer remain unanswered and unchanged. There is still no support for the suggestion that this is an assembly intermediate, and I am not convinced by the revised evolutionary analysis that Psak is unique. When considered together with the medium resolution and map quality, the low abundance of this tetrameric form and the fact that the abundance of this PSI form positively correlates with detergent levels (albeit not from the current authors work), I regret to say that I cannot support the publication of this work in *Nature* communications.

Reviewer #1 (Remarks to the Author):

Although I think the manuscript is much improved, the authors have not addressed my main concern. As I stated previously, “To prove relevance, the authors must show that the tetrameric state of PSI is present in *Cyanophora* in vivo.”

My main criticism is that there is no data suggesting that tetramers are present in vivo, and there is data that shows correlation between DDM and this unique tetramer state. The most straightforward interpretation is that the tetrameric state the authors have isolated is not physiologically relevant, and is instead induced by DDM.

The authors state that 0.1% DDM is very low for solubilization, which I agree with, but the standard procedure is to solubilize at high DDM concentration, separate the protein target, and then bring the DDM concentration to ~0.02%, which is around 2 times the CMC of DDM. Here, the authors solubilize the sample at 0.1% and then maintain the 0.1% which is around 20 times the CMC of DDM. Why this deviation from standard practice? I assume this is because 0.02% DDM results in too few tetramers for cryo-EM which would be consistent with the results from Watanabe that show a decrease in tetramers with lower DDM concentration.

Author reply:

First of all, we dramatically improved our revised manuscript according to your comments. We tested detergent concentration for solubilization of thylakoids, and added a new Figure to the revised manuscript (Supplementary Fig. 21, which shows results of trehalose gradient centrifugation of 0.1% DDM and 1% DDM solubilized thylakoid membranes). Also, to examine the functional significance of the PSI tetramer, we measured time-resolved fluorescence which showed some differences to that of the *Anabaena* PSI tetramer. We added discussions regarding the fluorescence differences between *Cyanophora* PSI tetramer and *Anabaena* PSI tetramer.

We revised the method of preparation of PSI tetramers. In the previous manuscript, it was described that 0.1% DDM was used to solubilize the thylakoid membranes, but actually it was 1%. We sincerely apologize for the wrong description of detergent concentration, and modified the method section in the revised manuscript accordingly.

We tested a lower concentration of DDM (0.1%) for solubilization of thylakoids. Because the efficiency of solubilization is very low by 0.1% DDM, the solubilized fractions were concentrated with Amicon, and then subjected to trehalose gradient

centrifugation. The result shows that even with the treatment of 0.1% DDM, the tetramer band was clearly obtained (Supplementary Fig. 21), and ratio of PSI monomer to PSI tetramer was not much changed between 0.1% and 1% DDM solubilization. This strongly indicates that *Cyanophora* has PSI tetramer *in vivo*. Further experiment to examine the presence or absence of the PSI tetramer *in vivo* is not possible at present, and it may be examined further by *in situ* cryo-electron tomography in the future.

The 0.1% DDM contained in the trehalose buffer for trehalose gradient centrifugation is required for the purity of PSI tetramer, because lower concentration DDM, e.g. 0.02%-0.03%, results in contamination of proteins other than PSI. Thus, the 0.1% DDM in the buffer is used and necessary for preparation of the PSI tetramer from *Cyanophora*.

Reviewer #2 (Remarks to the Author):

My concerns regarding the functionality and relevance of the PSI tetramer remain unanswered and unchanged. There is still no support for the suggestion that this is an assembly intermediate, and I am not convinced by the revised evolutionary analysis that Psak is unique. When considered together with the medium resolution and map quality, the low abundance of this tetrameric form and the fact that the abundance of this PSI form positively correlates with detergent levels (albeit not from the current authors work), I regret to say that I cannot support the publication of this work in Nature communications.

Author reply:

First of all, we dramatically improved our revised manuscript according to your comments. We tested detergent concentration for solubilization of thylakoids, and added a new Figure to the revised manuscript (Supplementary Fig. 21, which shows results of trehalose gradient centrifugation of 0.1% DDM and 1% DDM solubilized thylakoid membranes). Also, to examine the functional significance of the PSI tetramer, we measured time-resolved fluorescence which showed some differences to that of the *Anabaena* PSI tetramer. We added discussions regarding the fluorescence differences between *Cyanophora* PSI tetramer and *Anabaena* PSI tetramer.

We revised the method of preparation of PSI tetramers. In the previous manuscript, it was described that 0.1% DDM was used to solubilize the thylakoid membranes, but actually it was 1%. We sincerely apologize for the wrong description of detergent concentration, and modified the method section in the revised manuscript accordingly.

We tested a lower concentration of DDM (0.1%) for solubilization of thylakoids. Because the efficiency of solubilization is very low by 0.1% DDM, the solubilized fractions were concentrated with Amicon, and then subjected to trehalose gradient centrifugation. The result shows that even with the treatment of 0.1% DDM, the tetramer band was clearly obtained (Supplementary Fig. 21), and ratio of PSI monomer to PSI tetramer was not much changed between 0.1% and 1% DDM solubilization. This strongly indicates that *Cyanophora* has PSI tetramer *in vivo*. Further experiment to examine the presence or absence of the PSI tetramer *in vivo* is not possible at present, and it may be examined further by *in situ* cryo-electron tomography in the future.

Reviewers' Comments:

Reviewer #1:

Remarks to the Author:

The manuscript is improved as described by the authors in their rebuttal. Although I am still not entirely convinced that tetrameric PSI is found in Cyanophora, but the authors at least left this as a possibility to be investigated by future experiments.

My only outstanding concerns are regarding citations. The authors have over-cited themselves and neglected other's contributions to understanding the structures of PSI from different cyanobacteria. For example, in the Introduction, lines 44-47, the authors have cited 18 papers here to support the statement that structural organization of PSI varies, 11 of which are from their group. I think it would be more appropriate to cite a few key papers and/or reviews. Why cite the author's far-red light photosystem I structure? If there is a good reason, there are at least two other FRL-PSI structures from different cyanobacterial species. If the authors are going to cite every PSI paper structure, they are missing quite a few. For example, the recent PSI tetramer structure <https://doi.org/10.1016/j.xplc.2021.100248> and a recent PSI structure from high-light tolerant cyanobacteria <https://doi.org/10.7554/eLife.67518>.

As another example, on line 123 of the Results, it appears that the authors are citing all of the PSI structure papers. If this is the case, they have missed many, none of which are from their group, which is disappointing. Here is a list of papers that should additionally be cited:

- <https://doi.org/10.1016/j.crstbi.2020.08.003>
- <https://doi.org/10.1038/s41477-018-0130-0>
- <https://doi.org/10.1016/j.jbc.2021.101408>
- <https://doi.org/10.1038/s41477-020-0593-7>
- <https://doi.org/10.7554/eLife.67518>
- <https://doi.org/10.1016/j.xplc.2021.100248>
- <https://doi.org/10.1126/sciadv.aay6415>
- <https://doi.org/10.1038/s41467-019-12955-3>
- <https://doi.org/10.1038/s41598-021-00236-3>

I have the same concerns about over self-citations and neglecting contributions from others on lines 160 and 269. Please consider adding these citations and scouring the literature for other appropriate citations.

Finally, in the Discussion, lines 371-372, please edit this sentence. Reference 36, Koike et al., suggest that the Cyanophora PSI is only in monomeric form. Reference 5 suggests that it is in tetrameric, dimeric, and monomeric forms, where the tetramer and dimer decrease with lower detergent concentration.

Reviewer #2:

Remarks to the Author:

This new version of the manuscript includes an additional section regarding the red states of PSI which was not included before. The authors refer to a publication that was not peer reviewed yet. I do not know the N.comm policy on this issue, but in my opinion this is not appropriate, especially as I have some concerns regarding some of the experimental results shown in ref 22 (bioRxiv DOI: <https://doi.org/10.1101/2021.09.29.462462>).

The authors state that there is no 730 nm emission Cyanophora, I am not sure they are correct in this statement and if they want to keep it, they should fit their 77K emission data to better describe the underlying states. Their low temp absorption spectrum clearly shows some red sites, which may or may not be a consequence of tetramerization. Again, to prove this they might performed some measurements on their monomeric sample. Given that the quality of the map in the sites they term

low1 and low2 and their reliance on data before it was peer reviewed, I think this part is better omitted from the manuscript, supported independently or the cited manuscript should at the very least be accepted for publication.

The additional data supplied is sufficient to show that solubilization alone is probably not the cause of tetramerization. What is still missing is some data on the number of biological replicas (independent growth experiments prior to PSI purification) this experiment was performed. This should be clearly stated in the text and shown in the supplementary data section, when this information is provided, I think the concerns regarding the native state of PSI are reasonably addressed.

The manuscript still contains many language errors that should be removed before publication.

Reviewer #1 (Remarks to the Author):

The manuscript is improved as described by the authors in their rebuttal. Although I am still not entirely convinced that tetrameric PSI is found in Cyanophora, but the authors at least left this as a possibility to be investigated by future experiments.

First of all, we thank the reviewer for his/her positive and important comments and suggestions to improve our manuscript.

Comment 1:

My only outstanding concerns are regarding citations. The authors have over-cited themselves and neglected other's contributions to understanding the structures of PSI from different cyanobacteria. For example, in the Introduction, lines 44-47, the authors have cited 18 papers here to support the statement that structural organization of PSI varies, 11 of which are from their group. I think it would be more appropriate to cite a few key papers and/or reviews. Why cite the author's far-red light photosystem I structure? If there is a good reason, there are at least two other FRL-PSI structures from different cyanobacterial species. If the authors are going to cite every PSI paper structure, they are missing quite a few. For example, the recent PSI tetramer structure <https://doi.org/10.1016/j.xplc.2021.100248> and a recent PSI structure from high-light tolerant cyanobacteria <https://doi.org/10.7554/eLife.67518>.

Author reply 1:

Based on the reviewer's comments, we have deleted one ref. regarding the FRL-PSI structure, and added three new refs. from other groups, two of which were pointed out by the reviewer and another one is regarding the PSI tetramer structure from another group (ref. 16 of the revised manuscript). Furthermore, we added two reviews (ref. 4 and 6) regarding the composition and changes of PSI from prokaryotes to eukaryotes.

As another example, on line 123 of the Results, it appears that the authors are citing all of the PSI structure papers. If this is the case, they have missed many, none of which are from their group, which is disappointing. Here is a list of papers that should additionally be cited:

- <https://doi.org/10.1016/j.crstbi.2020.08.003>
- <https://doi.org/10.1038/s41477-018-0130-0>
- <https://doi.org/10.1016/j.jbc.2021.101408>

- <https://doi.org/10.1038/s41477-020-0593-7>
- <https://doi.org/10.7554/eLife.67518>
- <https://doi.org/10.1016/j.xplc.2021.100248>
- <https://doi.org/10.1126/sciadv.aay6415>
- <https://doi.org/10.1038/s41467-019-12955-3>
- <https://doi.org/10.1038/s41598-021-00236-3>

I have the same concerns about over self-citations and neglecting contributions from others on lines 160 and 269. Please consider adding these citations and scouring the literature for other appropriate citations.

Author reply 1:

According to the comments of the reviewer, we added most papers of the PSI structures pointed out by the reviewer, except those regarding the structures of FRL-PSI and PSI in complex with IsiA, as well as the method of PSI structural analysis by XFEL, as we thought that these papers focused on either the structures of chlorophyll f-binding (FRL-PSI), or IsiA (PSI-IsiA), or methodological development (XFEL), rather than the structure of PSI core itself. We also cited three critical reviews of PSI structures “Fromme et al., *Biochim. Biophys. Acta, Bioenerg.* **1507**, 5-31 (2001)”, “Suga and Shen, *Curr. Opin. Struct. Biol.* **63**, 10-17 (2020)”, “Hippler and Nelson, *Plant Cell Physiol.* **62**, 1073-1081 (2021)” in the revised manuscript. These three reviews show the evolution and variety of PSI structures from prokaryotes to eukaryotes, and are good for the summarization of our current understanding on PSI structural differences and evolution.

Comment 2:

Finally, in the Discussion, lines 371-372, please edit this sentence. Reference 36, Koike et al., suggest that the Cyanophora PSI is only in monomeric form. Reference 5 suggests that it is in tetrameric, dimeric, and monomeric forms, where the tetramer and dimer decrease with lower detergent concentration.

Author reply 2:

According to the comment of the reviewer, we largely modified the corresponding sentence in the revised manuscript to indicate that Koike et al. suggested that the Cyanophora PSI is only in monomeric form, whereas Reference 5 suggests that it can be found in tetrameric, dimeric and monomeric forms (lines 333-336).

Reviewer #2 (Remarks to the Author):

Comment 1:

This new version of the manuscript includes an additional section regarding the red states of PSI which was not included before. The authors refer to a publication that was not peer reviewed yet. I do not know the N.comm policy on this issue, but in my opinion this is not appropriate, especially as I have some concerns regarding some of the experimental results shown in ref 22 (bioRxiv DOI: <https://doi.org/10.1101/2021.09.29.462462>).

The authors state that there is no 730 nm emission Cyanophora, I am not sure they are correct in this statement and if they want to keep it, they should fit their 77K emission data to better describe the underlying states. Their low temp absorption spectrum clearly shows some red sites, which may or may not be a consequence of tetramerization. Again, to prove this they might performed some measurements on their monomeric sample. Given that the quality of the map in the sites they term low1 and low2 and their reliance on data before it was peer reviewed, I think this part is better omitted from the manuscript, supported independently or the cited manuscript should at the very least be accepted for publication.

Author reply 1:

First of all, we thank the reviewer for his/her important comments and suggestions to improve our manuscript. According to the comments of the review, we removed the section regarding the red states of PSI (Low1 and Low2) together with the citation of ref. 22 (bioRxiv DOI: <https://doi.org/10.1101/2021.09.29.462462>) in the revised manuscript.

Comment 2:

The additional data supplied is sufficient to show that solubilization alone is probably not the cause of tetramerization. What is still missing is some data on the number of biological replicas (independent growth experiments prior to PSI purification) this experiment was performed. This should be clearly stated in the text and shown in the supplementary data section, when this information is provided, I think the concerns regarding the native state of PSI are reasonably addressed.

Author reply 2:

According to the comments of the reviewer, we added biological replicates to the

Supporting Information in the revised manuscript.

Comment 3:

The manuscript still contains many language errors that should be removed before publication.

Author reply 3:

We have looked over the entire text and improved the writing where necessary.